# Not All Data are Good Labels:
# On the Self-supervised Labeling for Time Series Forecasting

**Yuxuan Yang[1,2], Dalin Zhang[3], Yuxuan Liang[4], Hua Lu[5], Gang Chen[1,2], Huan Li[1,2]** ✉

[1]The State Key Laboratory of Blockchain and Data Security, Zhejiang University
[2]Hangzhou High-Tech Zone (Binjiang) Institute of Blockchain and Data Security
[3]Space Information Research Institute, Hangzhou Dianzi University
[4]The Hong Kong University of Science and Technology (Guangzhou)
[5]Department of Computer Science, Aalborg University, Denmark
`{yangyuxuan, cg, lihuan.cs}@zju.edu.cn`
`zhangdalin@hdu.edu.cn, yuxliang@outlook.com, luhua@cs.aau.dk`

## Abstract

Time Series Forecasting (TSF) is a crucial task in various domains, yet existing TSF models rely heavily on high-quality data and insufficiently exploit all available data. This paper explores a novel self-supervised approach to re-label time series datasets by inherently constructing candidate datasets. During the optimization of a simple reconstruction network, intermediates are used as pseudo labels in a self-supervised paradigm, improving generalization for any predictor. We introduce the Self-Correction with Adaptive Mask (SCAM), which discards overfitted components and selectively replaces them with pseudo labels generated from reconstructions. Additionally, we incorporate Spectral Norm Regularization (SNR) to further suppress overfitting from a loss landscape perspective. Our experiments on eleven real-world datasets demonstrate that SCAM consistently improves the performance of various backbone models. This work offers a new perspective on constructing datasets and enhancing the generalization of TSF models through self-supervised learning. The code is available at https://github.com/SuDIS-ZJU/SCAM.

## 1 Introduction

Time Series Forecasting (TSF) is a crucial task with extensive applications in energy, finance, engineering, and many other domains. Recent advances in deep learning have resulted in TSF methods that outperform traditional methods in precision, robustness, and scalability [53, 31, 36].

Nevertheless, deep learning-based TSF methods still face significant challenges such as overfitting, dependence on high-quality datasets, and inconsistent performance across datasets — issues exacerbated by flawed evaluation practices [34, 30]. Central to these challenges is the problem of *low-quality labels*[1], associated with inherent noise and anomalies in raw data. Existing strategies, such as multimodal data integration [43, 4] and dataset scaling [35], focus on augmenting or refining datasets but fail to address the core limitation: the reliance on raw labels as ground truth. To address this, we pose two critical questions:

1. *Can the reliance on high-quality labeled time series datasets be alleviated, given their scarcity?*

---

[1]In TSF, given two adjacent windows in a time series, $x$ (input window) and $y$ (output window), "labels" refer to $y$ fitted by predictions $\hat{y} = f(x; \theta)$ in a supervised learning setting.

2. *Can the potential of existing time series datasets be better exploited to improve model performance?*

We posit that both answers are positive by redefining how labels are generated. *Instead of treating raw labels as immutable targets, we selectively replace them with "pseudo labels" generated self-supervisedly.*

The key idea is that the pseudo labels can be created from an inherent search through *candidate datasets* created by an auxiliary reconstruction task. In this process, raw labels are partially replaced with reconstructions, guided by an adaptive mask that identifies overfitted raw components and selectively replaces them with pseudo labels for predictions in the supervised setting. This self-supervised learning paradigm significantly enhances the generalization of TSF models compared to traditional supervised learning, which rigidly adheres to raw labels.

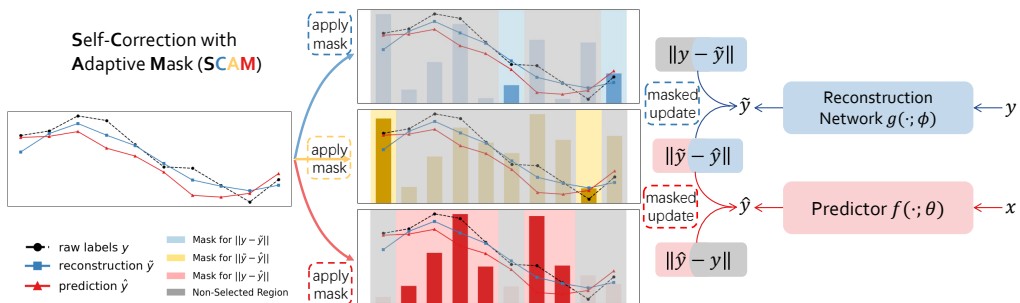

Figure 1: The illustration of the proposed method SCAM.

From an intuitive perspective, a straightforward reconstruction network $g(\cdot; \phi)$ can be employed to generate pseudo-labels in a self-supervised manner. To assess the feasibility of training such a reconstruction network, we conduct an initial evaluation using a grid search algorithm (Section 3.1, Appendix B). The network $g$ is optimized to learn an identity mapping (i.e., reconstruction) of the raw labels over multiple training epochs. During each epoch, the intermediate parameters of $g$ can be frozen to produce an approximate reconstruction of the dataset. Empirical observations reveal that when certain predictors $f(\cdot; \theta)$ are trained from scratch on these intermediate datasets, they exhibit superior performance compared to their counterparts trained exclusively on the raw datasets.

While the reconstruction-based approach demonstrates potential for pseudo-label generation, determining optimal parameters $\phi$ presents a significant challenge. Specifically, the reconstruction loss (i.e., $\|y - \tilde{y}\|$), when combined with the original supervised loss (i.e., $\|\hat{y} - y\|$), tends to drive $g(\cdot; \phi)$ toward reproducing the raw labels. This phenomenon induces an undesirable "overfitting" behavior during training.

To better analyze overfitting, we derive a mask form of the co-objective loss, which partitions original time series into distinct components (see Figure 1). We employ a criterion, namely the *sharpness metric* $\lambda_{max}$ [10], which detects overfitting by evaluating the sharpness of loss landscape. Using $\lambda_{max}$ and practical evaluations of the decomposed loss, we identify the overfitted components. This process, termed ***Self-Correction with Adaptive Mask*** (SCAM), discards raw data labels ($y$) based on reconstruction results ($\tilde{y}$) and current predictions ($\hat{y}$), smoothing the loss landscape and enhancing the generalization of the predictor model $f(\cdot; \theta)$.

We apply masked updates on a reconstruction network $g(\cdot; \phi)$ and a predictor model $f(\cdot; \theta)$ during training. During inference, the prediction will be generated directly by the updated $f(\cdot; \theta)$ **with no additional cost**. To further generalize across models of varying complexities, we also propose to apply Spectral Norm Regularization [46, 26] to parameters of the first or last linear layers without altering optimizers like previous works [10].

We summarize our contributions as follows:

- **Novel Perspective**: We explore a novel approach of self-enhancing TSF datasets by incorporating an auxiliary reconstruction task into TSF model training.

- **Self-supervised Paradigm**: We propose a self-supervised paradigm that generates pseudo labels from reconstructions and adaptively replaces overfitted raw labels to improve models' generalizability.

- **Detailed Analysis**: We confirm the effectiveness of the proposed self-supervised mask formulation with extensive analyses over various backbones and real-world datasets.

## 2 Preliminary

### 2.1 Problem Formulation

Many previous TSF studies adopt a paradigm that learns a direct mapping between two adjacent windows: the history series (*inputs*) and the future series (*labels*). Let the history series (*resp.* future series) be $\{\mathbf{x}_1, \mathbf{x}_2, \ldots, \mathbf{x}_N\} = X \in \mathbb{R}^{L \times N}$ (*resp.* $\{\mathbf{y}_1, \mathbf{y}_2, \ldots, \mathbf{y}_N\} = Y \in \mathbb{R}^{H \times N}$) with time series length $L$ (*resp.* $H$) and dataset size (number of segmented windows) $N$. For simplicity, we formulate the problem in the univariate scenario, as it naturally extends to the multivariate case by treating each variable as an additional dimension.

**Definition 1.** *A typical TSF process is formulated as a supervised-learning problem, i.e., to find* $\theta^* = \arg\min_\theta \|f(X; \theta) - Y\|$, *where a specified metric* $\|\cdot\|$ *is used to measure errors, typically the* $\ell_1$- *or* $\ell_2$-*norm.*

When splitting the data into training and test sets, the training set $D_{trn} = \{X_{trn}, Y_{trn}\}$ and the model $f(\cdot; \theta)$ can determine a minimal target error $\mathcal{L}_{tar} = \|f(X_{test}; \theta^*) - Y_{test}\|$ on the test set $D_{test} = \{X_{test}, Y_{test}\}$.

Usually, TSF models struggle on small or noisy datasets. Now, suppose we can obtain additional ***candidate datasets*** beyond the observed raw dataset; ideally, this would address the issue. In this sense, we assume a family of such candidate datasets $\mathcal{D} = \{D_i\}$, where the optimal candidate dataset $D_i^*$ enables better generalization for a predictor when evaluated by $\mathcal{L}_{tar}$ on the raw test set.

Without constraints, defining such a dataset family can be overly arbitrary. To handle this, we parameterize the family using a neural network $g(X; \phi) = \tilde{X}$, which is trained on a reconstruction loss $\mathcal{L}_{rec} = \|\tilde{X} - X\|$. During an iterative optimization process (e.g., a typical full-batch gradient descent with learning rate $\eta$), a candidate dataset $D_i$ is generated at each iteration step $i$ as follows:

$$D_i = \left\{ \tilde{X}_i, \tilde{Y}_i \right\} = \{\{g(x; \phi_i) \mid x \in X_i\}, \{g(y; \phi_i) \mid y \in Y_i\}\},$$

$$\phi_i = \phi_0 - \sum_{k=0}^{i-1} \sum_{x \in X_i \cup Y_i} \eta \nabla_{\phi_k} \|g(x; \phi_k) - x\|. \tag{1}$$

This approach shifts the focus from designing models with different inductive biases to identifying candidate datasets that improve a model's generalization. For TSF scenarios where data are often noisy and challenging to clean, replacing raw datasets with such parameterized candidates can lead to more robust performance. To conclude, we formalize the idea as follows.

**Hypothesis 1.** *Given time series data split into* $D_{trn} = \{X_{trn}, Y_{trn}\}$ *and* $D_{test} = \{X_{test}, Y_{test}\}$, *and a predictor model* $f(\cdot; \theta)$. *In a family of training sets* $\mathcal{D}_\phi = \{D_{trn}^i\}$ *parameterized by* $g(\cdot; \phi)$ *as in Eq. 1, there exist an optimal* $D^* = \{\tilde{X}^*, \tilde{Y}^*\} = \{\{g(x_j; \phi^*)\}, \{g(y_j; \phi^*)\}\}$ *such that*

$$\|f(X_{test}; \theta^*(\phi^*)) - Y_{test}\| \leq \|f(X_{test}; \theta^*(\phi_i)) - Y_{test}\|, \forall \phi_i,$$

*where* $\theta^*(\phi_i)$ *indicates that* $\theta^*$ *is optimized on* $D_{trn}^i = \{\tilde{X}_i, \tilde{Y}_i\} = \{\{g(x_j; \phi_i)\}, \{g(y_j; \phi_i)\}\}$.

### 2.2 Proposed $g(\cdot; \phi)$ for Reconstruction

We proceed to introduce a simple reconstruction network used in subsequent exploration. Similar to a predictor model, the reconstruction network operates in a sequence-to-sequence fashion, learning a function $g(\cdot; \phi)$ that maps raw series $Y$ to reconstructed series $\tilde{Y}$. Note that reconstruction is applied only to $Y$, time series datasets are typically generated from a single series using a moving window approach, where $X$ and $Y$ are almost fully overlapped. By skipping reconstruction for $X$, the predictor model can use raw series as inputs, avoiding *extra* inference costs or potential distribution shifts.

As depicted in Figure 2, the proposed $g(\cdot; \phi)$ comprises four convolutional layers, a cross-layer concatenation for multi-resolution integration, and a lightweight feedforward network (FFN) to decode the reconstructed results. The convolutional layers primarily act as a parameterized smoothing mechanism, similar to techniques for seasonal-trend decomposition [48, 16, 17]. The FFN then mixes information from different positions in a time series to reconstruct each data point using features extracted by convolutional layers (further details in Appendix D). Without introducing extra noise or patch-wise/point-wise masks [27], we directly learn an identity mapping.

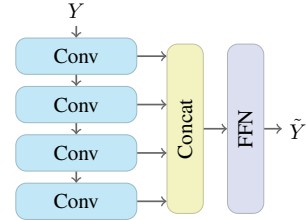

Figure 2: Convolution-FFN reconstruction network.

In essence, $g(\cdot; \phi)$ is designed to traverse diverse alternatives with varying levels of fidelity, which differs in purpose from the usual reconstruction task [22, 24, 18]. Due to this difference, the architecture of $g(\cdot; \phi)$ might benefit from novel designs; however, this is not examined in this study. The proposed $g(\cdot; \phi)$ is merely a simple prototype to validate our ideas and is not claimed superior to other unexplored options for this novel task.

## 3 Method

### 3.1 Initial Case

We begin with a simple case where $g(\cdot; \phi)$ evolves from a randomly initialized state toward an approximation of the raw target series $Y$. This process can be viewed as a *grid search* along the axis of the following reconstruction loss:

$$\ell_{rec} = \|g(y; \phi) - y\|. \tag{2}$$

In this setup, we optimize $g(\cdot; \phi)$ for reconstruction loss using full-batch gradient descent, while the predictor $f(\cdot; \theta)$ is optimized with mini-batch SGD in a vanilla training setting. At each optimization epoch $i$, we freeze the current parameters of $g(\cdot; \phi_i)$ and generate a candidate dataset $D_i$. The predictor $f(\cdot; \theta)$ is re-initialized before the epoch and trained on $D_i$ until convergence. The prediction results on the original test set are recorded for each $g(\cdot; \phi_i)$ and corresponding trained $f(\cdot; \theta_i^*)$. A case experiment is conducted on the ETTh1 dataset using a vanilla 2-layer MLP model (for faster and more guaranteed convergence). The pseudo-code for this grid search is provided in Appendix B.

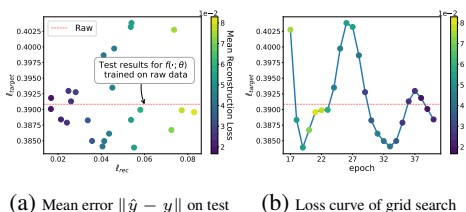

(a) Mean error $\|\hat{y} - y\|$ on test    (b) Loss curve of grid search

Figure 3: (a) shows the distribution of candidate datasets during the grid search. (b) shows the unsmooth loss curve in grid search. Each scatter in both figures denotes a converged predictor $f(\cdot; \theta)$ on an individual dataset parameterized by $g(\cdot; \phi_i)$.

So far, we can construct a series of candidate datasets $\{D_i\}$ through a two-step optimization process, each easily distinguished by its reconstruction loss, which intuitively measures its similarity to the raw dataset. Evaluating the performance of the same predictor trained on these varying candidate datasets leads to three major observations:

**Observation 1** (Improved Labels Enhance Performance). *Figure 3(a) demonstrates the existence of better-labeled candidates which bring better performance for a given predictor, as indicated by the scatter points below the red dotted line — which represents the baseline performance of the predictor trained on the raw dataset.*

**Observation 2** (Variable Performance w.r.t. Reconstruction Metric $\ell_{rec}$). *A feasible method should evaluate or rank candidate datasets during training without relying on test set performance. However, this remains challenging as datasets with comparable $\ell_{rec}$ values frequently exhibit substantial differences in actual performance (see Figure 3(a)).*

**Observation 3** (Difficult Training). *Training involves an* unstable *loss curve (Figure 3(b)), meaning many potentially superior candidate datasets (or equivalently $g(\cdot; \phi)$) could be missed. Moreover, training is costly. To ensure the predictor converges, $\phi$ is only updated after $\theta$ is fully trained. This renders grid search impractical for more complex models (e.g., PATCHTST and iTRANSFORMER) or larger datasets.*

In the next section, we propose replacing the brute-force grid search algorithm with a *co-objective training* approach that improves training stability and overall performance.

## 3.2 Co-objective Training

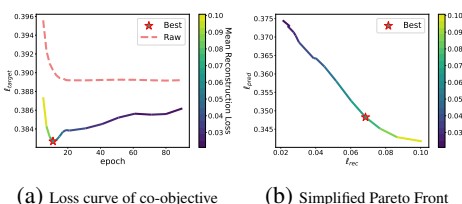

(a) Loss curve of co-objective    (b) Simplified Pareto Front

Figure 4: (a) shows the overfitting tendency supported by the increasing $\ell_{target}$ which measures errors on the test set. (b) shows the Pareto Front of two objectives, tilting top-left indicating the overfitting in (a).

The grid search (Section 3.1) is framed as a two-step optimization process with two distinct objectives involved in finding optimal candidate datasets. The reconstruction optimization primarily provides a trajectory of parameters $\phi_i$, without emphasizing optimality. In contrast, the prediction optimization evaluates the predictor's actual performance.

Our analysis of grid search results suggests that simplifying the training process into a **co-objective optimization** would be beneficial. Since the solution of two-step optimization (though not optimal on the test set) essentially lies on the Pareto Front of the corresponding co-objective optimization (see Figure 4(b)), a natural approach is to search along this front. Again, the trajectory of the optimization, rather than its strict optimality, contributes to improved test set performance, the co-objective training can still facilitate the construction of effective candidate datasets.

A single-step optimization using mini-batch SGD would be sufficient, enabling a more smooth trajectory of $\phi$ during updates (Figure 5(b) vs 5(a) and Figure 4(a) vs Figure 3(b)). Moreover, enabling gradient computation of $\ell_{pred} = \|\tilde{y} - \hat{y}\|$ w.r.t. $\phi$ introduces a regularization effect, making $\tilde{y}$ updated more cautiously towards $y$. This allows us to constrain the update of candidate datasets (now specifically represented by the reconstructed $\tilde{y}$) by jointly constraining gradients w.r.t. both $\theta$ and $\phi$:

$$\begin{aligned} \underset{\theta,\phi}{\text{minimize}} \quad & \mathcal{L} = \|\tilde{y} - y\| + \|\tilde{y} - \hat{y}\| \\ \text{subject to} \quad & \tilde{y} = g(y; \phi), \quad \hat{y} = f(x; \theta), \\ & \|\nabla_{\theta,\phi}\tilde{y}_i\| \le \delta, \quad \forall \tilde{y}_i \in \tilde{Y} \end{aligned} \quad (3)$$

where $\phi$ is trained using both loss terms whereas $\theta$ is trained solely on $\ell_{pred} = \|\tilde{y} - \hat{y}\|$. A gradient constraint $\delta$ is added to ensure a *smooth* trajectory of $\tilde{y}$, enabling the co-objective to traverse potential candidate datasets more carefully.

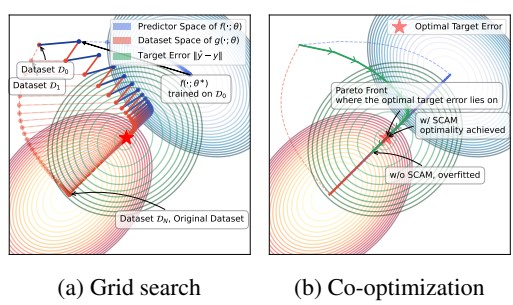

(a) Grid search    (b) Co-optimization

Figure 5: Illustrations of grid search and co-optimization.

The gradient constraint has a surrogate $\|\nabla_\theta f\|$ and practically implemented by Spectral Norm Regularization (SNR), as discussed in Section 3.4 and Appendix C. For simplicity, we temporarily omit this constraint, as a 2-layer MLP converges quickly with a small $\|\nabla_\theta f\|$. Using the same setting as the grid search, we evaluate the revised loss function for co-optimizing the predictor and the reconstruction network. Figure 4 shows that this co-training improves $\ell_{target}$ loss while simplifying the two-step training process, leading to more stable optimization and reduced training costs.

However, as the co-training process progresses, it becomes increasingly prone to overfitting (see Figure 4(a)). Overfitting is a fundamental issue in machine learning, tied to the generalizability of models. In this specific case, this issue arises as the reconstructed dataset gradually approaches the raw dataset, causing the target loss to converge to those of the raw dataset. Similar to the two-step grid search, determining a *reasonable threshold* to identify optimal parameters remains a challenge.

To address this specific overfitting issue, we propose solutions summarized in two main components of our method: Self-Correction with Adaptive Mask (SCAM) in Section 3.3 and Spectral Norm Regularization (SNR) in Section 3.4.

## 3.3 Self-Correction with Adaptive Mask (SCAM)

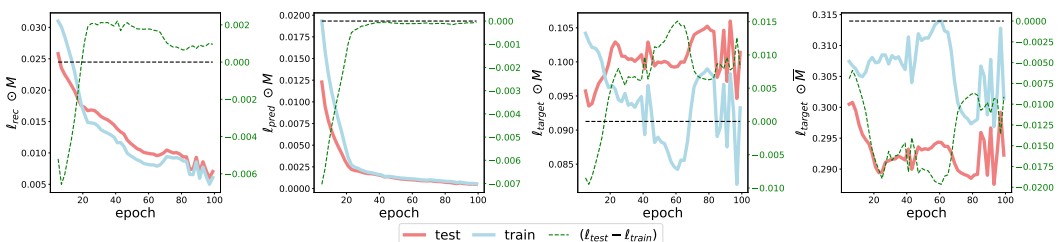

Figure 6: Loss curves of 4 parts divided by the adaptive mask. From left to right, $2|\tilde{y} - y| \odot \overline{M_<} \odot M$, $2|\tilde{y} - \hat{y}| \odot M_< \odot M$, $|y - \hat{y}| \odot M$ and $|y - \hat{y}| \odot \overline{M}$. The four parts combined are the loss $\mathcal{L}$ in Eq. 4. $M$ refers to the mask where $m_i > 0$ and $\overline{M}$ refers to the mask where $m_i < 0$. The right y-axis (in green) indicates the value for $(\ell_{test} - \ell_{train})$, the differences in losses measured on the test set and the train set. Values (on right y-axis) above 0 (the black dotted line) indicate overfitting.

**Mask Form of Self-supervised Loss**. In a traditional supervised-learning paradigm, the target loss $\ell_{target} = \|\hat{y} - y\|$ is used only to train a model, implying that all data points are equally treated as valid labels. However, in our approach, where predictors are trained alongside a search for candidate datasets guided by the reconstruction loss, the labels perceived by the predictors are adaptively shifted. Specifically, $\hat{y} = f(x; \theta)$ is trained to fit $\tilde{y}$, meaning only the second term is optimized for the predictor $f(\cdot; \theta)$ (Eq. 3). The reconstruction loss term, on the other hand, is optimized to provide self-supervised labels for the predictor. We frame this as a self-supervised-learning paradigm that *adaptively* adjusts labels in TSF problems. By comparing the revised loss with the traditional supervised loss, we can explicitly separate the auxiliary loss from the supervised loss: When revisiting the objective $\mathcal{L}$ in co-training, the additional loss term $\mathcal{L}_{aux}$ does not directly contribute to the target objective. Instead, this term depends on the relative positions of $\hat{y}$, $\tilde{y}$ and $y$. When the reconstructed $\tilde{y}$ is viewed as *a correction of labels*, $\mathcal{L}_{aux}$ indicates where the correction should be placed. Time series are naturally sparse in real scenarios, often containing spiking signals due to irregular events or anomalies. $\mathcal{L}_{aux}$ encourages the reconstructed $\tilde{y}$ to lie between the prediction $\hat{y}$ and the actual labels $y$, which can undermine sparsity when used as labels.

$$\mathcal{L} = \underbrace{|y - \hat{y}|}_{\mathcal{L}_{sup}} + \underbrace{(|\tilde{y} - \hat{y}| + |\tilde{y} - y| - |y - \hat{y}|)}_{\mathcal{L}_{aux}}$$

$$\text{Let } A = \tilde{y} - \hat{y}, \ B = \tilde{y} - y,$$

$$\mathcal{L} = \mathcal{L}_{sup} + (|A| + |B| - |A - B|)$$

$$= \mathcal{L}_{sup} + \begin{cases} 2\min\{|A|, |B|\}, & \text{if } AB > 0, \\ 0, & \text{if } AB \leq 0 \end{cases} \quad (4)$$

$$\text{Let } m = (\tilde{y} - \hat{y})(\tilde{y} - y),$$

$$\mathcal{L} = \mathcal{L}_{sup} + \begin{cases} 2\min\{|\tilde{y} - \hat{y}|, |\tilde{y} - y|\}, & \text{if } m > 0, \\ 0, & \text{if } m \leq 0 \end{cases}$$

$$= \mathcal{L}_{sup} + 2\left(|\tilde{y} - \hat{y}| \odot M_< + |\tilde{y} - y| \odot \overline{M_<}\right) \odot M.$$

Here, $M$ is a binary mask defined by $m = (\tilde{y} - \hat{y})(\tilde{y} - y) > 0$, $M_<$ ensures $|\tilde{y} - \hat{y}| < |\tilde{y} - y|$, and $\overline{M_<}$ represents its complement.

**Decoupling Overfitted Components by Adaptive Masks**. From the above derivation, we obtain a mask-based co-training loss, allowing us to analyze the causes of overfitting via the mask. As described for $\mathcal{L}_{aux}$ in Eq. 4, the mask $M$ defines the relative positions of $y$, $\tilde{y}$, and $\hat{y}$. Specifically, $M$ masks $\ell_{pred}$ and $\ell_{rec}$ to zero when $m_i = (\tilde{y}_i - \hat{y}_i)(\tilde{y}_i - y_i) < 0$. Similarly, $\ell_{target} = |\hat{y} - y|$ can

also be divided using this mask. By comparing train/test differences in the divided losses (particularly $\ell_{target}$, since $\ell_{pred}$ and $\ell_{rec}$ are zero when $m_i < 0$), we identify that overfitting primarily stems from $\ell_{target} \odot M$, as shown in Figure 6.

Further evidence is provided by analyzing the loss landscape. A sharpness metric for optimization, proposed by [10], measures the generalization capability of a model. Specifically, the sharpness metric is defined as $\lambda_{max} = \|\nabla_\theta^2 \mathcal{L}\|_2^2$, the largest eigenvalue of the Hessian matrix. A higher $\lambda_{max}$ indicates a sharper loss landscape, which correlates with a worse generalization or more severe overfitting. By computing this metric on the converged parameters, we observe that $\ell_{target} \odot M$ does exhibit a sharper landscape compared to $\ell_{target} \odot \overline{M}$. To address this, we keep the less sharp part of the $\mathcal{L}_{sup}$ term, i.e., $\mathcal{L}_{sup} \odot \overline{M}$, resulting in the revised loss:

$$\mathcal{L} = \underbrace{\|y - \hat{y}\| \odot \overline{M}}_{\text{masked } \mathcal{L}_{sup}} + \underbrace{2\left(\|\tilde{y} - \hat{y}\| \odot M_< + \|\tilde{y} - y\| \odot \overline{M_<}\right) \odot M}_{\mathcal{L}_{aux}}. \tag{5}$$

Training with this revised loss reduces the sharpness of the loss landscape. As shown in Figure 7, this effectively mitigates overfitting in the co-training scenario. This represents the final loss form in our proposed method, termed Self-Correction with Adaptive Mask (SCAM), where the mask $M$ is constructed based on an auxiliary reconstruction task. The adaptive $M$ effectively identifies and removes overfitted components of time series labels, enabling the search for more validated parametric candidate datasets $g(y; \phi_i)$.

### 3.4 Spectral Norm Regularization (SNR)

Note that we have omitted the gradient constraints in Eq. 3. This term is, in fact, positively correlated with $\nabla_\theta f(x; \theta)$ (discussed in Appendix E). This further supports our previous analysis, as we have only tested MLP models, which converge easily with lower $\nabla_\theta f$. However, when the predictor is replaced by a *Transformer-based* model, the dominant source of overfitting shifts from $\ell_{target}$ to $\ell_{pred}$. This phenomenon is not unique to our self-supervised learning paradigm. For example, [10] proposed sharpness-aware optimization to address overfitting in supervised settings. While gradient penalties are theoretically effective, they may not be practical for complex models like Transformers, as computing second-order derivatives can significantly hinder optimization.

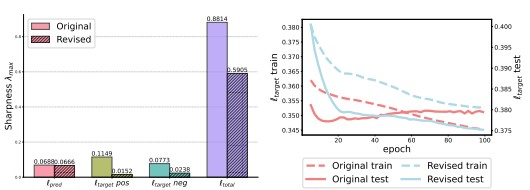

(a) Sharpness comparison  (b) Loss curve comparison

Figure 7: Effectiveness of the revised loss form

A more direct approach is to regularize parameters using the sharpness metric, known as *Spectral Norm Regularization* [26, 46]:

$$W_{normalized} = \gamma \cdot \frac{W}{\|W\|_2}, \tag{6}$$

where $\|\cdot\|_2$ is the spectral norm (the largest eigenvalue of parameter matrix) and $\gamma$ is a learnable scale factor. When applying to self-attention (SA) in Transformer-based architecture, SNR significantly undermines the expressiveness of attention score matrices (entropy collapse [49]). Hence, Ilbert et al. [10] conclude that SNR is inapplicable to SA parameters. However, we observe that linear layers — typically the embedding layer before SA and the projection layer after SA — also contribute to the overall sharpness of the loss landscape. Consequently, we propose applying SNR selectively to the pre- and post-SA linear layers. In this way, SNR can work with MLP-based models as they are also composed of multiple linear layers. Further empirical studies on SNR are discussed in Section E.3.

## 4 Experiment and Analysis

We address three major questions for experiments:

- **Q1**: Is SCAM effective across different backbone models and datasets with varying features?
- **Q2**: How do SCAM and SNR contribute to the potential improvement in model performance?
- **Q3**: How does the self-supervised reconstruction task benefit the predictor models?

Table 1: Performance boost by adding SCAM and SNR to different backbones. Better results are in **bold**.

| Models | MLP | | + Ours | | CYCLENET | | + Ours | | PATCHTST | | + Ours | | iTRANS | | + Ours | |
|---|---|---|---|---|---|---|---|---|---|---|---|---|---|---|---|---|
| Metric | MSE | MAE | MSE | MAE | MSE | MAE | MSE | MAE | MSE | MAE | MSE | MAE | MSE | MAE | MSE | MAE |
| ETTh1 | 0.464 | 0.448 | **0.437** | **0.433** | 0.457 | 0.441 | **0.431** | **0.429** | 0.469 | 0.455 | **0.427** | **0.433** | 0.454 | 0.448 | **0.431** | **0.440** |
| ETTh2 | 0.382 | 0.405 | **0.366** | **0.394** | 0.388 | 0.409 | **0.362** | **0.393** | 0.387 | 0.407 | **0.370** | **0.398** | 0.383 | 0.407 | **0.377** | **0.402** |
| ETTm1 | 0.391 | 0.402 | **0.388** | **0.398** | 0.379 | 0.396 | **0.368** | **0.388** | 0.387 | 0.400 | **0.381** | **0.394** | 0.407 | 0.410 | **0.387** | **0.399** |
| ETTm2 | 0.280 | 0.325 | **0.276** | **0.322** | 0.266 | 0.341 | **0.262** | **0.309** | 0.281 | 0.326 | **0.281** | **0.326** | 0.288 | 0.332 | **0.283** | **0.327** |
| Electricity | 0.204 | 0.285 | **0.203** | **0.283** | 0.168 | 0.259 | **0.166** | **0.258** | 0.205 | 0.290 | **0.191** | **0.275** | 0.178 | 0.270 | **0.173** | **0.267** |
| Traffic | 0.522 | 0.335 | **0.494** | **0.308** | 0.472 | 0.301 | **0.448** | **0.290** | 0.481 | 0.304 | **0.455** | **0.288** | 0.428 | 0.282 | **0.411** | **0.266** |
| Weather | 0.262 | 0.281 | **0.258** | **0.278** | 0.243 | 0.271 | **0.242** | **0.268** | 0.259 | 0.281 | **0.253** | **0.275** | 0.258 | 0.278 | **0.257** | **0.278** |

Our main evaluation involves seven datasets: Electricity, Weather, Traffic, and four ETT datasets (ETTh1, ETTh2, ETTm1, ETTm2), all of which are well-established TSF benchmarks and publicly available [40]. We also test the proposals using four PeMS datasets of a larger scale, as reported in Appendix H. The predictor (i.e., the backbone model integrated with SCAM) covers representative TSF models, including both MLP-based and Transformer-based architectures. MLP [15] is a vanilla 2-layer baseline equipped with RevIN [12] while CYCLENET [17] is a SOTA MLP-based model explicitly capturing cyclic trend in time series. PATCHTST [27] and iTRANSFORMER [21] are Transformer-based models, representing channel-independent and channel-dependent methods, respectively. Following previous settings [53, 40, 41] for direct and fair comparison, we set prediction length $H \in \{96, 192, 336, 720\}$ and look-back length to 96 for all datasets. We provide dataset descriptions, implementation details, and reproduction instructions in Appendix G.

## 4.1 Main Experiment (Q1)

Table 1 demonstrates consistent performance improvements in all backbones across all datasets when SCAM and SNR are incorporated in the self-supervised-learning paradigm. The full results with detailed breakdowns by prediction lengths are provided in Appendix H. These gains are particularly notable on ETT datasets, which are known for their noisy nature and relatively small size. Notably, Transformer-based models like PATCHTST and iTRANS, which typically underperform compared to lightweight models MLP and CYCLENET on these datasets, show significant enhancements in generalization with SCAM. On the Weather dataset, the boost is more modest, likely due to the intrinsic chaotic nature of atmospheric data.

Regarding **Q1**, our method demonstrates general effectiveness across various backbones and datasets. A well-known discrepancy between MLP-based and Transformer-based models is their dataset preferences: Transformer-based methods excel on large, regular datasets, while MLP-based methods perform better on noisy datasets [30, 34]. SCAM helps bridge this gap by enabling Transformer-based models to perform competitively on traditionally challenging datasets and enhancing the robustness of MLP-based models.

## 4.2 Ablation Study (Q2)

Table 2: Ablation study for CYCLENET.

| +SCAM | +SNR | ETTh1 | | ETTh2 | | ETTm1 | | ETTm2 | | Weather | |
|---|---|---|---|---|---|---|---|---|---|---|---|
| | | MSE | MAE | MSE | MAE | MSE | MAE | MSE | MAE | MSE | MAE |
| ✗ | ✗ | 0.444 | 0.436 | 0.381 | 0.407 | 0.379 | 0.397 | 0.266 | 0.313 | 0.248 | 0.273 |
| ✗ | ✓ | 0.438 | 0.432 | 0.372 | 0.398 | 0.375 | 0.392 | 0.265 | 0.311 | 0.246 | 0.272 |
| ✓ | ✗ | 0.436 | 0.432 | 0.365 | 0.395 | 0.371 | 0.390 | 0.263 | 0.311 | 0.242 | 0.270 |
| ✓ | ✓ | **0.431** | **0.429** | **0.362** | **0.393** | **0.368** | **0.388** | **0.262** | **0.309** | **0.242** | **0.268** |

Table 3: Ablation study for iTRANS.

| +SCAM | +SNR | ETTh1 | | ETTh2 | | ETTm1 | | ETTm2 | | Weather | |
|---|---|---|---|---|---|---|---|---|---|---|---|
| | | MSE | MAE | MSE | MAE | MSE | MAE | MSE | MAE | MSE | MAE |
| ✗ | ✗ | 0.438 | 0.440 | 0.400 | 0.417 | 0.413 | 0.414 | 0.299 | 0.341 | 0.285 | 0.306 |
| ✗ | ✓ | 0.443 | 0.445 | 0.399 | 0.417 | 0.413 | 0.415 | 0.306 | 0.347 | 0.283 | 0.302 |
| ✓ | ✗ | 0.436 | **0.438** | 0.381 | 0.404 | 0.391 | 0.404 | 0.293 | 0.342 | 0.274 | 0.289 |
| ✓ | ✓ | **0.431** | 0.440 | **0.377** | **0.402** | **0.387** | **0.399** | **0.283** | **0.327** | **0.257** | **0.278** |

To answer **Q2**, we present the performance gains from SCAM and SNR through an ablation study (Tables 2 and 3 [2]), using iTRANS (Transformer-based) and CYCLENET (MLP-based) as representatives (the SOTA one in their category).

SNR, a practical alternative to the gradient penalty in Eq. 8, consistently enhances performance across backbones. However, iTRANS, being more prone to overfitting on small datasets (discussed in [10]),

---

[2] The results in both tables may have discrepancy in baseline results from Table 1. This is because the Table 1 has baseline results from the original papers, and Table 2 and 3 are from our runs for module modification. See Sec G.3 for further explanations.

benefits less from SNR compared to CYCLENET, likely due to insufficient data for generalization in higher-complexity architectures. With SCAM, both models exhibit significant improvements, leveraging the expanded effective training data.

In summary, SCAM is the primary driver of performance gains, offering consistent and clear improvements. While SNR enhances results as a standalone method, it serves best as a complement to SCAM, acting as an effective surrogate for the gradient penalty in the self-supervised objective.

### 4.3 SCAM: A Multiple Instance Learning View (Q3)

Multiple Instance Learning (MIL [25, 3]) is a classical weakly-supervised binary classification problem where typically a bag of instances is labeled positive if at least one instance is positive. [8, 5] extend MIL to Time Series Classification by treating an input window as a bag of instances (time points), enabling a model to predict based on instance-level classifications that are more interpretable.

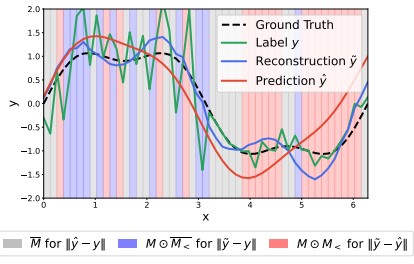

Figure 8: Case study of visualized mask.

To answer **Q3**, we hypothesize that the effectiveness of SCAM shares similarities with MIL by leveraging instance-level signals, though it does not strictly follow a MIL framework. For a quick illustration, we synthesize a toy dataset where the ground truth is defined as $y = A\sin(\omega_1 x) + B\sin(\omega_2 x)$, with added noise sampled alternately from $\mathcal{N}(0, \sigma_1)$ and $\mathcal{N}(0, \sigma_2)$ in different windows. As shown in Figure 8, when the noise deviation $\sigma$ is large (left part), SCAM tends to optimize $\ell_{rec} = M \odot \overline{M_<}\|\tilde{y} - y\|$, prioritizing robust reconstructions; when $\sigma$ is small, SCAM shifts focus to optimizing $\ell_{pred} = M \odot M_<\|\tilde{y} - y\|$ and $\ell_{target} = \|\hat{y} - y\|$, emphasizing accurate predictions.

This study only reveals a part of SCAM's self-supervision effectiveness, which is further explored in Appendix E.

## 5 Related Work

We discuss related techniques from meta-learning and self-supervised learning perspectives, with an inventory of TSF models in Appendix F.

**Meta-Learning for Time Series**. Meta-Learning, by definition, seeks to perform in a learn-to-learn paradigm. Generally speaking, meta-learning for TSF includes optimization-based and metric-based methods. Optimization-based methods target optimal initial parameters [9], often involving a two-loop optimization process [28, 38]. Metric-based methods [7, 38] learn a metric function that provides more expressive measurements of distances between data samples, commonly comparing training and validation samples.

Our method SCAM aligns with the broader scope of meta-learning. Specifically, the grid search (Section 3.1, Appendix B) follows the two-loop structure similar to optimization-based methods. However, it diverges by focusing on *dataset space* rather than *parameter space*. Additionally, the final mask form of SCAM in essence provides a more accurate metric tailored for a supervised-learning setting. Previous works [38, 7] learn metrics on the sample level (a window of time series), whereas ours focuses on the instance level (individual data points of time series).

**Self-supervised Learning for Time Series**. Self-supervised learning trains models without relying on manually labeled data by using auxiliary tasks like generation, contrast, and reconstruction to learn expressive representation or create pseudo labels. In the realm of time series, this approach is discussed more in Time Series Classification (TSC) [11, 42, 22]. Recent works [8, 5] present a novel perspective that instances in time series/segments can have multiple labels. They propose corresponding weakly-supervised-learning methods that significantly improve both performance and interpretability.

In this work, we integrate the *multiple-label* perspective by employing an auxiliary reconstruction task, commonly used in TSC, to enhance the performance of TSF. The pseudo labels, often discussed in TSC, are derived from existing, manual ones, while ours are created in a self-supervised paradigm.

# 6    Conclusion and Future Work

This paper presents a self-supervised approach SCAM that enhances TSF models by selectively replacing overfitted components with pseudo labels derived from intermediate reconstructions. When combined with Spectral Norm Regularization, SCAM improves generalization and robustness across TSF models. Future work will explore extending SCAM to tasks such as time series outlier detection and error correction.

# 7    Acknowledgment

This work was supported by the National Natural Science Foundation of China (Nos. 62402420, 62402414), Fundamental Research Funds for Central Universities (Grant No. 226202400109), the Zhejiang Provincial Natural Science Foundation (No. LD24F020015), and the National Regional Innovation and Development Joint Fund (No. U24A20254). Huan Li is the corresponding author of this paper.

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

## A   Limitations

Our work constrains the discussion for TSF models in MLP- and Transformer-based ones, excluding other potentially feasible solutions [23, 1]. We omit these newly explored architectures since they have not outperformed the baseline methods we have adopted under the same budget. Also, we have not verified our method on LLM-based models (or models of an LLM-level scale) [39, 2] due to the excessive costs required.

## B   Grid Search Algorithm in Initial Case

In Section 3.1, we have introduced the grid search process to illustrate how the reconstruction network $g(\cdot; \phi)$ evolves to approximate the raw target series $Y$ by minimizing the reconstruction loss $\ell_{rec} = \|g(y; \phi) - y\|$. The following Algorithm 1 provides a detailed implementation of this process, where the reconstruction network $g(\cdot; \phi)$ and the predictor model $f(\cdot; \theta)$ are jointly optimized to minimize both reconstruction and prediction losses. Specifically, for each candidate reconstruction parameter $\phi_i$ (line 1), the predictor's parameters $\theta$ and its optimizer are initialized (line 2). In the inner loop (line 3), the predictor is optimized by first reconstructing $\tilde{y}_i$ using $g(\cdot; \phi_i)$ (line 6), then calculating the prediction loss $\ell_{pred}$ (line 7) and reconstruction loss $\ell_{rec}$ (line 8). The prediction loss $\ell_{pred}$ is backpropagated to update $\theta$ (line 8), and this process repeats until the gradient $\nabla_\theta$ falls below a threshold $\alpha$ or the maximum steps $J$ are reached (line 3). After optimizing $\theta$, the reconstruction loss $\ell_{rec}$ is backpropagated to update $\phi$ (line 11), and this process is repeated for all $N$ candidates to find the best $\phi$ (line 1).

---

**Algorithm 1** Grid Search along $\ell_{rec}$

---

**Parameters :** $\phi$ w.r.t reconstruction network $g(\cdot; \phi)$;
$\qquad\qquad\quad$ $\theta$ w.r.t predictor model $f(\cdot; \theta)$
**Optimizers :** $Opt_\phi$ w.r.t $\phi$ (outer loop);
$\qquad\qquad\quad$ $Opt_\theta$ w.r.t $\theta$ (inner loop)
**Initialize** $\quad$ : $\phi, Opt_\phi$

1 **for** $i \leftarrow 0$ **to** $N$ **do**
$\qquad\qquad$ ▷ $N$ for number of candidates
2 $\quad$ Initialize $\theta, Opt_\theta$; $j \leftarrow 0$
3 $\quad$ **while** $\nabla_\theta > \alpha$ **and** $j \leq J$ **do**
$\qquad\qquad$ ▷ $J$ for maximum steps to optimize a predictor
4 $\quad\quad$ $\ell_{rec} \leftarrow 0$
5 $\quad\quad$ **foreach** $(x, y) \in (X, Y)$ **do**
6 $\quad\quad\quad$ $\tilde{y}_i \leftarrow g(y; \phi_i)$
7 $\quad\quad\quad$ $\ell_{pred} \leftarrow \|f(x; \theta) - \tilde{y}_i\|$
8 $\quad\quad\quad$ $\ell_{rec} \leftarrow \ell_{rec} + \|\tilde{y}_i - y\|$
9 $\quad\quad\quad$ Backpropagate $\ell_{pred}$ and update $\theta$ using $Opt_\theta$
10 $\quad\quad$ $j \leftarrow j + 1$
11 $\quad$ Backpropagate $\ell_{rec}$ and update $\phi$ using $Opt_\phi$

---

## C   Further Discussion on Gradient Constraint

Recall that in Section 3.2, we have proposed the co-training objective as:

$$\underset{\theta, \phi}{\text{minimize}} \quad \mathcal{L} = \|\tilde{y} - y\| + \|\tilde{y} - \hat{y}\|$$

$$\text{subject to} \quad \tilde{y} = g(y; \phi), \quad \hat{y} = f(x; \theta),$$

$$\|\nabla_{\theta, \phi} \tilde{y}_i\| \leq \delta, \quad \forall \tilde{y}_i \in \tilde{Y}.$$

With constraints on the gradient of $\tilde{y}$, the co-training update allows for a more stable optimization compared to the grid search using a two-step optimization. When viewing each intermediate $\phi_i$ as an individual candidate dataset, applying cautious updates to $\tilde{y}_i = g(y; \phi_i)$ introduces additional candidate datasets along the optimization trajectory, enriching the search process.

However, since $\nabla_{\theta,\phi}\tilde{y}_i$ is not practically computable for each $\tilde{y}_i$ within a single step of optimization, we turn to look for a surrogate term to replace this constraint.

Note that

$$
\begin{aligned}
\nabla_{\phi,\theta}\tilde{y}_i &= [\nabla_\phi g \quad \nabla_\theta g] \\[2mm]
&= \left[\nabla_\phi g \quad \nabla_\phi g \frac{\nabla_\theta \mathcal{L}}{\nabla_\phi \mathcal{L}}\right] \\[2mm]
&= \left[\nabla_\phi g \quad \nabla_\phi g \frac{\nabla_\theta\left[(\tilde{y}-y)^2+(\tilde{y}-\hat{y})^2\right]}{\nabla_\phi\left[(\tilde{y}-y)^2+(\tilde{y}-\hat{y})^2\right]}\right] \\[2mm]
&= \left[\nabla_\phi g \quad \nabla_\phi g \frac{2(\hat{y}-\tilde{y})\nabla_\theta f}{2(\tilde{y}-y)\nabla_\phi g + 2(\tilde{y}-\hat{y})\nabla_\phi g}\right] \\[2mm]
&= \left[\nabla_\phi g \quad \frac{(\hat{y}-\tilde{y})\nabla_\theta f}{(\tilde{y}-y)+(\tilde{y}-\hat{y})}\right] \\[2mm]
&\approx [\nabla_\phi g \quad -\nabla_\theta f], \quad \text{because } |\tilde{y}-y| \ll |\tilde{y}-\hat{y}|.
\end{aligned}
\tag{7}
$$

Reconstruction is inherently a simpler task compared to forecasting, which allows the last approximation in Eq. 7 to hold after just a few steps of initial optimization. When $\tilde{y}$ is far distinct from original labels $y$, the reconstructed series $\tilde{y}$ becomes nearly unpredictable, leading to instability in the optimization process. Therefore, adding a constraint on $g$ can interfere with the convergence of the predictor model. To address this, Eq. 7 offers an optional assurance of gradient constraint, which uses $\|\nabla_\theta f\| \leq \delta$ as a surrogate for maintaining stability during optimization.

The constrained form of the optimization is equivalent to the penalized form using *Lagrangian Duality*. Eq. 3 can be rewritten as:

$$
\begin{aligned}
\underset{\theta,\phi}{\text{minimize}} \quad \mathcal{L} &= \|g(y;\phi)-y\| + \|g(y;\phi)-f(x;\theta)\| \\
&\quad + \beta\|\nabla_\theta f(x;\theta)\|.
\end{aligned}
\tag{8}
$$

For readers who are familiar with Reinforcement Learning, this derivation resembles the transfer from a constrained optimization to a penalized one (e.g., from TRPO [32] to PPO [33]). In brief, while the penalized form is theoretically equivalent to the constrained form, it is challenging to choose a fixed $\beta$ that works universally across all datasets or even within a single dataset (because intrinsic characteristics can vary over time). Thus, a more general form of constraint is required to better serve the penalty, similar to the concept of gradient clipping in PPO.

Note that $\nabla_\theta f \leq \delta$ implies the *Lipchitz condition* for an arbitrary function $f$. This means

$$
\|f(x_1;\theta)-f(x_2;\theta)\| \leq C(\theta)\|x_1-x_2\|,
\tag{9}
$$

where $C(\theta)$ is a constant with respect to the parameter $\theta$. When considering a typical Fully Connected Layer defined as $f(x;W,b) = \sigma(Wx+b)$, the condition becomes

$$
\begin{aligned}
&\|\sigma(Wx_1)-\sigma(Wx_2)\| \\
\approx\ &\|(\sigma(x_0)+\sigma'(x_0)(Wx_1-x_0))-(\sigma(x_0)+\sigma'(x_0)(Wx_2-x_0))\|, \quad \text{by Taylor expansion} \\
=\ &\|\sigma'(x_0)(W(x_1-x_2))\| \\
\leq\ &\sigma'_{max}\|W\|\|x_1-x-2\| \\
\leq\ &C(W)\|x_1-x_2\|.
\end{aligned}
\tag{10}
$$

When the derivative of activation function $\sigma$ has an upper bound $\sigma'_{max}$(as is often the case for common activation functions like ReLU, Sigmoid, etc.), the Lipchitz condition holds as long as

$$
\|W(x_1-x_2)\| \leq C(W,b)\|x_1-x_2\|.
\tag{11}
$$

We expect the constant $C$ to be relatively small so that the penalty works. In fact, $C$ here corresponds to the spectral norm of the matrix $W$, which is defined as

$$
\|W\|_2 = \max_{x\neq 0} \frac{\|Wx\|}{\|x\|}.
\tag{12}
$$

By applying the Spectral Norm Regularization (SNR) in Eq. 6, we can ensure the constant $C$ equals to exactly 1.

However, in practice, SNR have limitations when applied to the parameter matrix in self-attention mechanisms. This phenomenon is termed *entropy catastrophe* as discussed by [10]. In this paper, by analyzing the sharpness of different components in the predictor model, we propose to use pre-SNR and post-SNR combined, which specifically normalizes the first and last linear layer in the TSF models (see Section 3.4).

# D   Implementation Details of $g(\cdot; \phi)$

As introduced in Section 2.2, we propose a simple enough reconstruction network $g(\cdot; \phi)$ that serves our objective. Despite its simplicity, the architecture incorporates some special designs that enhance its performance. Specifically, these include the *conv-concat layer* and the *point-wise FFN*, which are detailed in Appendix D.1 and Appendix D.2, respectively.

## D.1   Conv-concat Layer

**Transpose and Unfold**.  The implementation details of the conv-concat layer involve two key operations that are designed for the following two benefits:

1. The convolution outputs can be concatenated into embeddings of the same length, enabling features from different frequencies to be ideally fused into one embedding.

2. The features are evenly arranged along the temporal dimension, ensuring that each embedding in the sequence has the same large Receptive Field.

To achieve these benefits, we introduce a two-step operation: **Transpose** and **Unfold**, which work together to ensure both uniform embedding structure and large Receptive Fields.

Specifically, we set kernel size $= 3$, stride $= 2$, padding $= 1$, and the number of kernels doubled for each subsequent layer. Using this setup, as illustrated in Figure 9, we ensure that the number of features remains invariant across different layers, with only the shape of the features changing. Now we can fuse the outputs from different convolution layers together by flattening/unfolding the features to the original shape of $(L \times 1)$. Again, considering the effectiveness of point-wise FFN presented in Appendix D.2, we expect the concatenated features to be near-equally arranged along the temporal dimension to preserve the sequential relationships in the embedding. To achieve this, we first transpose the features and then unfold them. This practice can ensure a Receptive Field of $(2^{l+1} - 1)$ wide for each embedding, where $l$ is the total number of convolution layers.

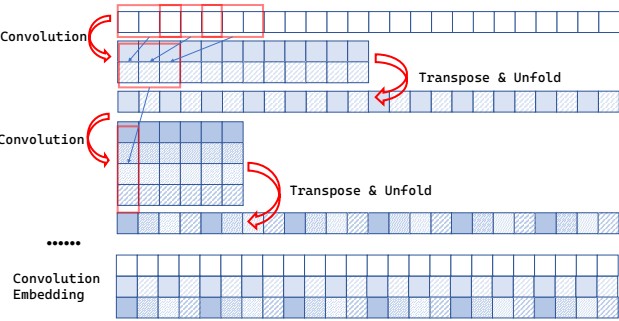

Figure 9: The illustration of transpose and unfold operation in the Convolution Encoder.

**Effective Receptive Field**.  As what was proposed by [23], the Effective Receptive Field (ERF) is a reasonable consideration for designing convolution-based architectures. To evaluate the ERF of our conv-concat layer, we input an impulse function and visualize the resulting ERF, as shown in Figure 10. The visualization demonstrates that, without requiring an extra-large convolution kernel, our proposed method achieves a near-global ERF. This is made possible by combining the outputs from different layers, each capturing distinct frequency patterns.

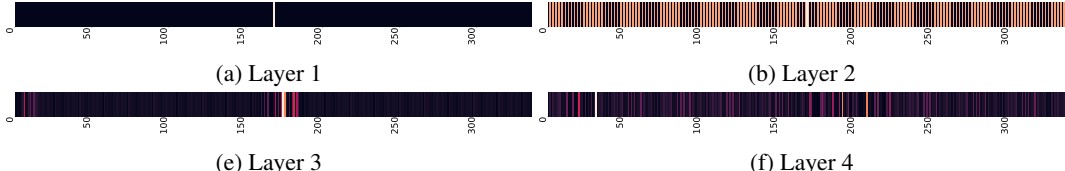

(a) Layer 1                               (b) Layer 2

(e) Layer 3                               (f) Layer 4

Figure 10: Effective Receptive Field (ERF) of the proposed conv-concat layer. By the transpose and unfold operation, the ERF of each convolution layer covers the entire input series with different frequencies.

## D.2 Point-wise FFN

We employ a point-wise FFN as a parameter-sharing module to decode the outputs from the convolution layer. The FFN is essentially a two-layer MLP that resembles the common design of a linear projector in predictor models, as mentioned in Section E.3.

To further illustrate, we present the three different parameterizations of the linear projector commonly used in TSF models in Figure 11:

- **Patch-dependent Design**: PATCHTST [27] adopts a patch-dependent linear projector (shown in Figure 11(a)) that first flattens all features within patches and utilizes an extra-large weight matrix of shape $Ld \times d$, where $L$ is the sequence length and $d$ is the dimension of the latent embedding.

- **Patch-independent Design**: [14] proposes that the necessity for patch-dependent designs depends on the specific task. For instance, tasks like forecasting may require patch-dependent projectors, while tasks like contrastive learning might favor a patch-independent design (shown in Figure 11(b)).

- **Point-wise Design**: Our reconstruction process does not require exploiting the patch correlations as a necessity and can even extend this independence to point-wise scope (shown in Figure 11(c)). This approach is feasible only when each point-wise embedding is sufficiently rich in information, a property achieved through our convolution layer, which provides a near-global ERF for each point.

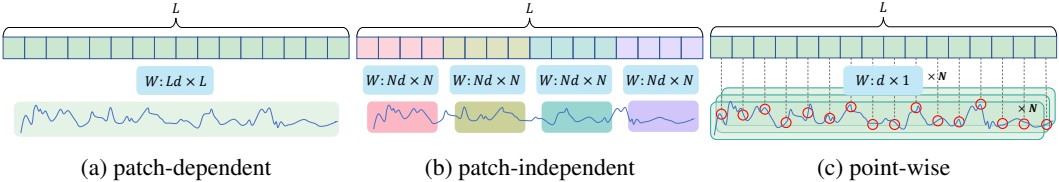

(a) patch-dependent             (b) patch-independent             (c) point-wise

Figure 11: Three types of linear projectors.

The sharing of parameters in our linear projector allows for an increase in parallel modules. In practice, we generate multiple reconstructed samples for the same raw input sample and feed them to the predictor simultaneously. This approach inherently expands the scale of the dataset, further enhancing training efficiency.

## E Further Analysis

### E.1 Distribution of Reconstructed Datasets in Grid Search

The distribution of sampled predictions is shown in Figure 12. The predictions are evaluated using three loss metrics: (1) **Prediction loss**: $\ell_{pred} = \|\hat{y} - \tilde{y}\|$, (2) **Reconstruction loss**: $\ell_{rec} = \|\tilde{y} - y\|$, and (3) **Target loss**: $\ell_{target} = \|\hat{y} - y\|$. In Figure 12, scatter points illustrate the relationships among these losses across various candidate datasets generated by $g(\cdot; \phi)$ on the ETTh1 dataset. Each scatter point is colored according to the mean reconstruction loss ($\ell_{rec}$) of its corresponding dataset. Lighter colors (e.g., yellow) represent datasets with higher $\ell_{rec}$, while darker colors (e.g., purple) correspond to datasets with lower $\ell_{rec}$.

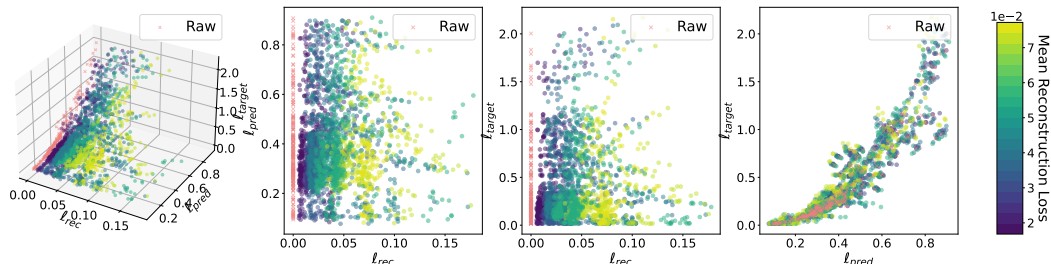

Figure 12: Distribution of losses of candidate datasets generated by $g(\cdot; \phi)$ on ETTh1.

The distribution is visualized across three projections:

1. $\ell_{rec}$-$\ell_{pred}$ *Plane* illustrates the relationship between the reconstruction loss $\ell_{rec}$ and the prediction loss $\ell_{pred}$, both of which are actively optimized during training. As $\ell_{rec}$ decreases (darker colors), the points in the distribution become more condensed, indicating reduced flexibility in the candidate datasets. This trend suggests that datasets with very low reconstruction loss may lack the diversity needed for optimal predictor performance.

2. $\ell_{rec}$-$\ell_{target}$ *Plane* highlights the relationship between the reconstruction loss $\ell_{rec}$ and the target loss $\ell_{target}$, where $\ell_{target}$ serves as the primary evaluation metric for predictor performance. Interestingly, datasets closer to the raw data (darker colors, with lower $\ell_{rec}$) do not consistently lead to better $\ell_{target}$ values. This observation, which indicates that overly strict reconstruction constraints may hinder prediction quality, is further supported by Figure 3.

3. $\ell_{pred}$-$\ell_{target}$ *Plane* directly examines prediction performance, as $\hat{y}$ is involved in calculating both $\ell_{pred}$ and $\ell_{target}$. Notably, candidate datasets with intermediate distances from the raw data (green points) demonstrate better generalization. These datasets are characterized by relatively higher $\ell_{pred}$ and lower $\ell_{target}$ and are distributed more toward the bottom-right region of the plane, reflecting improved prediction quality.

These results suggest that datasets with moderate reconstruction loss — neither too high nor too low — strike a better balance between flexibility and generalization. This balance ultimately leads to improved predictor performance, as overly strict reconstruction constraints may limit model adaptability, while overly high reconstruction loss may fail to capture meaningful patterns.s

### E.2 Demystifying the Self-supervision in SCAM

**Conv-concat Layers as Feature Amplifier**. In theory, the reconstruction network can be extended to larger scales as it does not interfere with inference efficiency. However, in our implementation, it is kept as simple as possible to prioritize training efficiency. Despite its simplicity, our observations reveal that the two different components of the reconstruction network function in distinct ways, which offers insights into designing more effective reconstruction networks.

As mentioned in Appendix D.2, the FFN in $g(\cdot; \phi)$ is designed point-wise, decoding the concatenated outputs of the convolution layers for each point independently. To further investigate the utility of conv-concat layers, we skip the initial linear layers and activation functions in $g(\cdot; \phi)$, directly applying the final linear transformation to the latent outputs of the conv-concat layers. This process generates an intermediate series, which we term as undecided $\tilde{y}'$.

Figure 13 shows that the $\tilde{y}'$ (red plots) effectively amplifies the sparse spiking signals present in the raw data. This behavior can be interpreted as the conv-concat layers acting as a **feature amplifier**, emphasizing and enlarging important patterns in the input data.

**Channel-wise Distribution Alignment**. Distribution shift, a core challenge in long-term Time Series Forecasting (TSF), is decisive for a TSF model to generalize on future data after training. The most widely adopted approach to address this issue is *Reversible Instance Normalization* (RevIn) [12, 20], which aligns the distributions of historical and future data.

While RevIn significantly improves the performance of TSF models, it falls short in aligning distributions *across channels in the multivariate forecasting setting*. Figure 14 highlights this limitation:

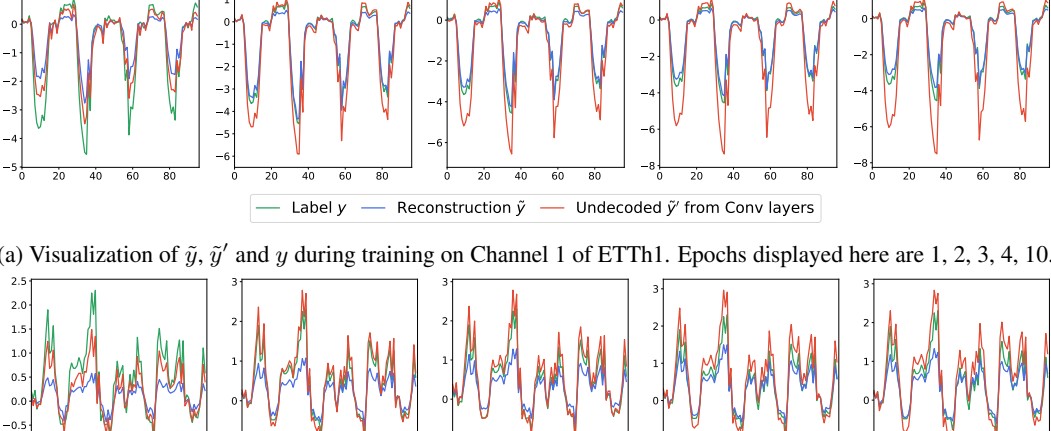

(a) Visualization of $\tilde{y}, \tilde{y}'$ and $y$ during training on Channel 1 of ETTh1. Epochs displayed here are 1, 2, 3, 4, 10.

(b) Visualization of $\tilde{y}$ and $\tilde{y}'$ during training and $y$ on Channel 4 of ETTh1. Epochs displayed here are 1, 2, 3, 4, 10.

Figure 13: Visualization our difference components in $g(\cdot; \phi)$.

even when the raw data $y$ is normalized by RevIn, the distribution distances (measured by the KL divergence metric) remain significant, as shown in the leftmost plots of all subfigures.

In this view, the proposed reconstruction network $g(\cdot; \theta)$ effectively serves as a general **channel-wise distribution alignment** mechanism. Interestingly, when examining the intermediate undecided $\tilde{y}'$ across channels, we observe that, although the sparse features are amplified, the distribution distances are reduced compared to those in the raw data. Overall, the final reconstructed series exhibits better alignment across channels. However, exceptions such as Channels 3 and 6 (see Figure 14(c)) demonstrate that the alignment is not uniform across all channels.

For models designed with *channel-independence* (CI), such as PATCHTST [27] and CYCLENET [17], aligning distributions across channels is of critical importance, especially when trained on datasets with a large number of variates. The channel-wise alignment introduced by our self-supervised reconstruction task provides an effective solution to distribution shifts, thereby enhancing the performance of CI predictor models.

**Evolution of Adaptive Mask in SCAM**. In Section 4.3, we have explored the benefits of the adaptive mask in SCAM from a Multiple Instance Learning (MIL) view, highlighting its sensitivity to instances with varying deviations and its ability to apply different masking strategies accordingly.

In this part, we further visualize the evolution process of the adaptive mask during training on real datasets, using ETTh1 as an example, as shown in Figure 15.

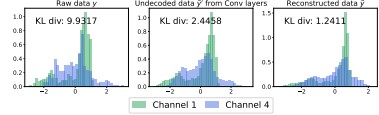
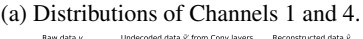

(a) Distributions of Channels 1 and 4.

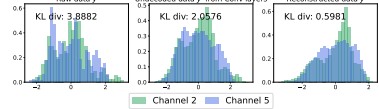

(b) Distributions of Channels 2 and 5.

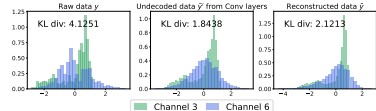

(c) Distributions of Channels 3 and 6.

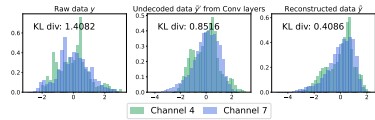

(d) Distributions of Channels 4 and 7.

Figure 14: Distribution alignment for channels in ETTh1 with data all normalized by RevIN.

At the very first epoch, the model heavily relies on the prediction $\hat{y}$ to reconstruct the series $\tilde{y}$. This benefits the convergence of both the predictor model $f(\cdot; \theta)$ and the reconstruction network $g(\cdot; \phi)$. As previously analyzed, the reconstructed series $\tilde{y}$ aligns the data distribution, enabling $f(\cdot; \theta)$ to learn effectively from $\tilde{y}$. Correspondingly, the prediction $\hat{y}$ in the first epoch is basically a draft with smooth curvature, preventing $\tilde{y}$ from quickly overfitting the noisy raw labels $y$. In subsequent epochs, the proportion of gray masks applied to the loss term $\|\hat{y}, y\|$

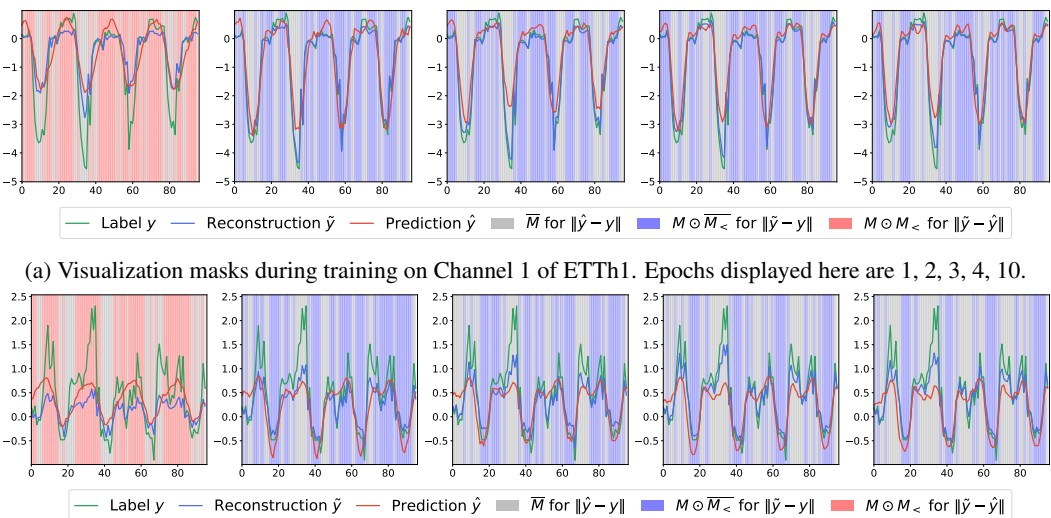

(a) Visualization masks during training on Channel 1 of ETTh1. Epochs displayed here are 1, 2, 3, 4, 10.

(b) Visualization masks during training on Channel 4 of ETTh1. Epochs displayed here are 1, 2, 3, 4, 10.

Figure 15: Evolution of the adaptive masks in SCAM during training.

increases. This indicates that the model progressively emphasizes a more precise fitting to the true labels, resulting in improved reconstruction and prediction quality.

In comparison with the case study in Section 4.3, the adaptive mask tends to collapse more rapidly from intermediate states where all three masking strategies are simultaneously active. This is possibly due to the noisy and unpredictable nature of real-world datasets. We plan to explore less aggressive strategies compared to binary masking, which hopefully will increase the robustness of the current method.

### E.3 In-depth Examination on SNR

We further verify the effectiveness of SNR by analyzing the loss landscape. Transformer-based models, as discussed, are more prone to overfitting, particularly on noisy datasets. This weakens our adaptive mask's ability to discard overfitted components, as $\ell_{pred}$ — the predictor's training loss — also contributes to overfitting. Gradient penalty offers a solution to this issue. For TSF models specifically, where both inputs and outputs are time series with abrupt distribution shifts, parameter robustness is critical for mitigating overfitting. Previous works [46, 26] have shown that regulating parameters via their spectral norm enhances stability against input perturbations, which is particularly beneficial in TSF.

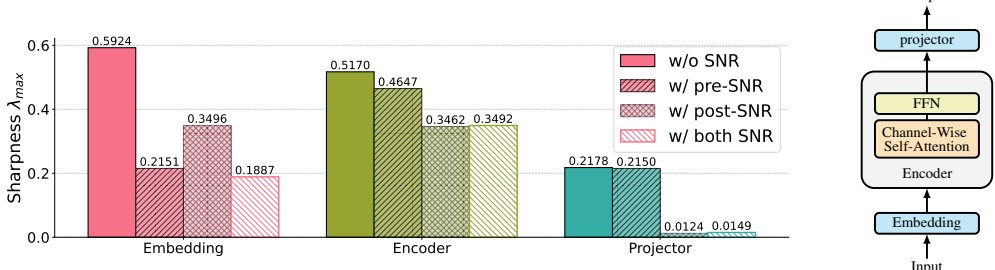

Figure 16: Sharpness of difference components in ITRANS.

Figure 17: ITRANS as a TST example

In Figure 16, the bars represent the sharpness of different components in ITRANS, divided into three parts: embedding, encoder, and projector (see Figure 17). The embedding and projector are linear layers, while the encoder comprises channel-wise attention blocks. This architectural pattern is

common among Time Series Transformers (TSTs), with the encoder being the primary focus and less emphasis placed on pre- or post-encoder linear layers.

Without SNR, the input linear layer (embedding) exhibits the highest sharpness, indicating it is the most overfitted component (see red bars). This suggests that overfitting in Transformers originates not only from self-attention mechanisms but also from the linear layers. By applying SNR to the pre- and post-encoder linear layers (Section 3.4), we observe smoother loss landscapes, validating SNR's ability to reduce sharpness effectively.

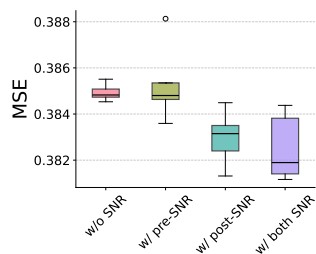

Figure 18: Empirical results on variants of SNR.

To determine the best practices for SNR, we conduct multiple runs with different random seeds to evaluate various design choices. As shown in Figure 18, applying SNR to the post-encoder linear layer or both linear layers proves most effective: 'post-SNR' achieves lower mean metrics and reduced variance, while 'both SNR' yields better mean performance but with higher variance. We recommend both practices, depending on the use case, and adopt 'both SNR' for all tests in this paper.

## F   Related Work Revisit

In this part, we will introduce some popularly adopted methods for TSF, in which our baseline models are included.

**Channel-independent Transformers**. Time Series Transformers (TST) [53, 40, 54] have recently led to significant progress in the TSF problem, demonstrating convincing superiority over traditional methods and convolution-based models. Inspired by [48], [27] incorporates the *Channel Independent* (CI) design (sharing weights across all channels) to introduce PATCHTST, a new state-of-the-art (SOTA) model that significantly benefits from CI and the patching operation. Multiple works follow this practice and achieve excellent performance in TSF [6, 19, 50].

**Channel-wise Transformers**. Building on TST, more recent research has focused on designing Transformers capable of capturing channel dependencies inherent in multivariate time series data. Notable examples include CROSSFORMER [51], iTRANSFORMER (iTRANS) [21], DSFORMER [47], and CARD [37]. These models shed light on exploiting inter-series relationships to improve forecasting accuracy. Furthermore, having observed that Time Series Transformers are inherently unstable due to a sharp loss scape, [10] propose a sharpness-aware optimizer to mitigate such issues. Their work focuses on an optimization-level approach involving a two-step backward. Nonetheless, channel-wise Transformers still suffer from overfitting on small datasets. By sharpness analysis, our proposed SCAM locates the overfitting issue and provides a solution from a data-level perspective.

**Linear or MLP-based Models**. In contrast to the quadratic complexity of Transformers, lightweight linear or MLP-based models have emerged as competitive alternatives offering simplicity and efficiency. RLINEAR and RMLP [15] verify that a vanilla linear model or a 2-layer MLP, when combined with a widely-adopted normalization method [12], can achieve near SOTA performance in TSF. Further research [52] on channel dependencies within linear and MLP-based models has yielded performance improvements over previous CI approaches. Moreover, the models [44, 45] that directly learn linear regression or MLP-based models on complex frequency features have achieved remarkable performances. Most recently, SPARSETSF [16], a highly lightweight model, incorporates 1D convolutions as down-sampling modules and learns linear parameters on the down-sampled values of the original series. CYCLENET [17], a SOTA model that explicitly captures periodic trend features to enhance vanilla linear or MLP-based models to be on par with Transformer-based models.

## G   Experiment Details

### G.1   Datasets

We conduct experiments on 11 real-world datasets to evaluate the performance of the proposed SCAM. The datasets are detailed below.

- **ETT** [53]: This dataset contains 7 factors of electricity transformers, recorded between July 2016 and July 2018. The subsets ETTh1 and ETTh2 are recorded hourly, while ETTm1 and ETTm2 are recorded every 15 minutes.

- **Electricity** [40]: This dataset records the hourly electricity consumption of 321 clients.

- **Traffic** [40]: This dataset collects hourly road occupancy rates measured by 862 sensors across the San Francisco Bay Area freeways, spanning from January 2015 to December 2016.

- **Weather** [40]: This dataset includes 21 meteorological factors, recorded every 10 minutes at the Weather Station of the Max Planck Biogeochemistry Institute in 2020.

- **PeMS**: This dataset contains public traffic network data from California, collected at 5-minute intervals. We use the same four subsets (PeMS03, PeMS04, PeMS07, PeMS08) as adopted in iTRANSFORMER (iTRANS) [21].

For the ETT datasets, we divide them by ratio $\{0.6, 0.2, 0.2\}$ into train set, validation set, and test set. For Electricity, Traffic, and Weather, we follow the same split ratio of $\{0.7, 0.1, 0.2\}$ as in TIMESNET [41, 53, 40]. For the PeMS datasets, we split them using the ratio $\{0.6, 0.2, 0.2\}$ following the same setting as iTRANS [21]. All datasets are scaled using the mean and variance of their respective training sets, a standard practice in TSF [41]. The statistics of all used datasets are listed in Table 4.

Table 4: Statistics of evaluation datasets.

| Datasets | ETTh1 | ETTh2 | ETTm1 | ETTm2 | Electricity | Traffic | Weather | PeMS03 | PeMS04 | PeMS07 | PeMS08 |
|---|---|---|---|---|---|---|---|---|---|---|---|
| # of TS Variates | 7 | 7 | 7 | 7 | 321 | 862 | 21 | 358 | 307 | 883 | 170 |
| TS Length | 17420 | 17420 | 69680 | 69680 | 26304 | 17544 | 52696 | 26209 | 16992 | 28224 | 17856 |

### G.2 Backbones

We evaluate the proposed SCAM against the following baseline backbone models:

1. MLP [15], a 2-layer MLP model combined with Reversible Instance Normalization [12].

2. CYCLENET [17], a 2-layer MLP equipped with efficient cycle modeling that belongs to a general seasonal-trend decomposition method (Cycle/MLP in the original paper).

3. iTRANS [21], a Transformer-based model that computes attention scores on the inverted series along the channel dimension.

4. PATCHTST [27], a Channel-Independent Transformer-based model that uses patching to tokenize the input.

It should be pointed out that the scale of a time series dataset is determined by a combination of the number of variates and the length of the dataset. Therefore, a fair comparison should take both factors into account. In our experiments, we observe that some baseline models, such as iTRANS, benefit greatly from datasets with a larger number of variates, while others, like PATCHTST, tend to perform better on longer datasets.

### G.3 Reproductivity

**Hyperparameters and Settings**. All experiments and methods are implemented in Python and PyTorch [29] and conducted on two Nvidia RTX A5000 Ada generation GPUs (32GB VRAMs) and two Nvidia RTX A6000 GPUs (48GB VRAMs). We use the ADAM optimizer [13] with a learning rate initialized as $\eta = 0.001$ for all settings. Unlike prior works [21, 16] that set a fixed, small number of training epochs, we adopt an early stopping strategy based on the MSE metric of the validation set, with a patience of 20 epochs.

For the reconstruction network $g(\cdot; \phi)$, the hyperparameters are detailed in Table 5.

We use 4 convolution layers in total, with consistent settings for kernel size, stride, and padding size as explained in Appendix D.1. Dim_multiplier represents the expansion ratio of convolution channels. The channels for the four convolution layers are set to 1, 2, 4, and 8, respectively, and are further multiplied by 4 to increase the model's capacity. Hidden_dim is the dimension size in the FFN, and # of series indicates that we use 8 point-wise linear projectors in parallel as explained in Appendix D.2

Table 5: Hyperparameters of the reconstruction network $g(\cdot; \phi)$.

| | |
|---|---|
| # of convolution layers | 4 |
| dim_multiplier | 4 |
| hidden_dim | 128 |
| # of series | 8 |

and Figure 11. The hyperparameter settings provided in Table 5 are applied consistently across all datasets.

**Rerun Baselines as Additional Comparisons.** To ensure fair comparisons with baseline results, we include both the results reported in the original papers and the results from our own re-implementations. Specifically: The MLP and CYCLENET results are taken from CYCLENET paper [17]. The PATCHTST and ITRANS results are from the ITRANS paper [21], as the PATCHTST paper [27] does not provide results with a look-back length of 96.

While the results from the original papers and our reruns show no major discrepancies, minor differences do exist. To provide complete transparency, we present both in Table 8 and Table 9 in Appendix H. Results from the original papers are more reliable as they reflect the authors' intended implementations, while our reruns ensure consistency in training settings for direct comparison.

**The Result Discrepancy on PeMS Datasets.** We also evaluate the larger traffic dataset, PeMS, which has been previously examined in both ITRANS and CYCLENET. However, we observe major discrepancies between our results and those reported in the original papers [21, 17].

While the original papers state that prediction lengths of $\{12, 24, 48, 96\}$ were used, their reported results closely align with what we obtain using prediction lengths of $\{12, 24, 36, 48\}$. The issue of reproduction inconsistencies is also widely discussed in the ITRANSFORMER GitHub issues. For reference, we provide the results obtained using our settings, which we hope will aid in clarifying these discrepancies.

## G.4 Efficiency

Table 6: The efficiency analysis of SCAM. All metrics are measured on the Electricity dataset with a batch size of 32.

| | # of parameters | Max Memory Allocated (GB) | Training Time (ms/iter) |
|---|---|---|---|
| MLP | 24.8K | 0.102 | 6.126 |
| +SCAM | 52.4K | 1.095 | 30.451 |
| CYCLENET | 78.7K | 0.099 | 6.879 |
| +SCAM | 106K | 1.092 | 30.832 |
| PATCHTST | 338K | 1.281 | 28.074 |
| +SCAM | 365K | 2.072 | 53.044 |
| ITRANS | 3.3M | 0.791 | 20.395 |
| +SCAM | 3.3M | 1.729 | 44.357 |

The SCAM method introduces an additional lightweight reconstruction network to aid the training. The additional computation cost is only introduced in training, because the additional module is discarded like a *scaffold*.

For the training efficiency, the baselines we adopted are acknowledged for their low cost. We only introduce a *constant* increase in memory and time consumption, enabling our method to be integrated into backbones that scale in parameters.

# H   More Experiment Results

In this section, we present detailed results from the previously mentioned experiments:

- **Tables 8 and 9**:
    - Summarize the main experiments on the ETT, Electricity, Traffic, and Weather datasets, with detailed breakdowns of different prediction lengths provided.
    - Baseline results in **Table 8** are taken from the original papers.
    - Baseline results in **Table 9** are reproduced by us.

- **Table 7**:
    - Reports experiments on the PeMS datasets, including baseline results from our runs.

- **Tables 10 and 11**:
    - Provide the complete results of the ablation studies.

In Table 7, 8 and 9, color RED indicates better performance and color BLUE indicates worse performance.

Table 7: Full results of experiments on PeMS datasets, comparing backbone models and their integration with our proposal. All baseline results are reproduced by us.

| Models | | MLP | | +Ours | | CYCLENET | | +Ours | | PATCHTST | | +Ours | | iTRANS | | +Ours |
|---|---|---|---|---|---|---|---|---|---|---|---|---|---|---|---|---|---|
| Metric | | MSE | MAE | MSE | MAE | MSE | MAE | MSE | MAE | MSE | MAE | MSE | MAE | MSE | MAE | MSE | MAE |
| PeMS03 | 12 | 0.083 | 0.191 | 0.082 | 0.189 | 0.073 | 0.179 | 0.072 | 0.178 | 0.078 | 0.186 | 0.072 | 0.176 | 0.075 | 0.186 | 0.074 | 0.180 |
| | 24 | 0.138 | 0.246 | 0.132 | 0.240 | 0.108 | 0.218 | 0.106 | 0.216 | 0.123 | 0.234 | 0.104 | 0.212 | 0.097 | 0.207 | 0.089 | 0.198 |
| | 36 | 0.196 | 0.297 | 0.189 | 0.291 | 0.147 | 0.256 | 0.144 | 0.255 | 0.172 | 0.276 | 0.136 | 0.244 | 0.127 | 0.237 | 0.130 | 0.242 |
| | 48 | 0.257 | 0.344 | 0.198 | 0.304 | 0.182 | 0.288 | 0.178 | 0.291 | 0.221 | 0.317 | 0.160 | 0.264 | 0.166 | 0.273 | 0.174 | 0.285 |
| | Avg. | 0.168 | 0.269 | 0.150 | 0.256 | 0.127 | 0.235 | 0.125 | 0.235 | 0.148 | 0.253 | 0.118 | 0.224 | 0.116 | 0.226 | 0.117 | 0.226 |
| PeMS04 | 12 | 0.103 | 0.211 | 0.103 | 0.211 | 0.092 | 0.198 | 0.091 | 0.197 | 0.101 | 0.208 | 0.084 | 0.190 | 0.084 | 0.188 | 0.080 | 0.183 |
| | 24 | 0.168 | 0.273 | 0.167 | 0.273 | 0.137 | 0.244 | 0.137 | 0.244 | 0.162 | 0.268 | 0.116 | 0.228 | 0.121 | 0.228 | 0.108 | 0.213 |
| | 36 | 0.246 | 0.335 | 0.243 | 0.333 | 0.187 | 0.289 | 0.187 | 0.289 | 0.227 | 0.321 | 0.147 | 0.261 | 0.151 | 0.257 | 0.139 | 0.244 |
| | 48 | 0.326 | 0.390 | 0.320 | 0.387 | 0.235 | 0.329 | 0.234 | 0.328 | 0.297 | 0.367 | 0.168 | 0.279 | 0.186 | 0.288 | 0.191 | 0.295 |
| | Avg. | 0.211 | 0.302 | 0.208 | 0.301 | 0.163 | 0.265 | 0.162 | 0.265 | 0.197 | 0.291 | 0.129 | 0.240 | 0.135 | 0.240 | 0.130 | 0.234 |
| PeMS07 | 12 | 0.079 | 0.185 | 0.080 | 0.185 | 0.069 | 0.171 | 0.069 | 0.171 | 0.076 | 0.180 | 0.068 | 0.169 | 0.063 | 0.159 | 0.060 | 0.154 |
| | 24 | 0.140 | 0.248 | 0.139 | 0.246 | 0.110 | 0.218 | 0.109 | 0.217 | 0.130 | 0.241 | 0.106 | 0.212 | 0.090 | 0.192 | 0.085 | 0.184 |
| | 36 | 0.210 | 0.306 | 0.209 | 0.304 | 0.153 | 0.260 | 0.152 | 0.260 | 0.184 | 0.286 | 0.144 | 0.248 | 0.135 | 0.242 | 0.132 | 0.237 |
| | 48 | 0.285 | 0.360 | 0.282 | 0.357 | 0.195 | 0.299 | 0.194 | 0.298 | 0.244 | 0.332 | 0.179 | 0.281 | 0.171 | 0.277 | 0.183 | 0.293 |
| | Avg. | 0.179 | 0.275 | 0.177 | 0.273 | 0.132 | 0.237 | 0.131 | 0.236 | 0.159 | 0.260 | 0.124 | 0.228 | 0.115 | 0.218 | 0.115 | 0.217 |
| PeMS08 | 12 | 0.093 | 0.198 | 0.094 | 0.199 | 0.082 | 0.184 | 0.082 | 0.184 | 0.087 | 0.191 | 0.130 | 0.187 | 0.077 | 0.176 | 0.071 | 0.167 |
| | 24 | 0.153 | 0.257 | 0.152 | 0.255 | 0.124 | 0.228 | 0.125 | 0.228 | 0.137 | 0.240 | 0.163 | 0.222 | 0.107 | 0.207 | 0.100 | 0.198 |
| | 36 | 0.223 | 0.312 | 0.220 | 0.310 | 0.170 | 0.268 | 0.170 | 0.267 | 0.194 | 0.291 | 0.163 | 0.222 | 0.142 | 0.239 | 0.130 | 0.223 |
| | 48 | 0.297 | 0.362 | 0.293 | 0.360 | 0.220 | 0.307 | 0.218 | 0.305 | 0.255 | 0.334 | 0.223 | 0.276 | 0.204 | 0.293 | 0.200 | 0.293 |
| | Avg. | 0.192 | 0.282 | 0.190 | 0.281 | 0.149 | 0.247 | 0.149 | 0.246 | 0.168 | 0.264 | 0.170 | 0.227 | 0.133 | 0.228 | 0.125 | 0.220 |
| Imp% | Avg. | - | - | 3.24% | 1.48% | - | - | 0.57% | 0.23% | - | - | 19.61% | 14.06% | - | - | 2.51% | 1.55% |

Table 8: Full results for performance comparisons between backbone models and their integration with our proposals on ETT, Electricity, Traffic, and Weather datasets. The results for MLP and CYCLENET are taken from the CYCLENET paper [27], while the results for PATCHTST and ITRANS are from the ITRANSFORMER paper [21].

| Models | | MLP | | +Ours | | CYCLENET | | +Ours | | PATCHTST | | +Ours | | ITRANS | | +Ours | |
|---|---|---|---|---|---|---|---|---|---|---|---|---|---|---|---|---|---|
| Metric | | MSE | MAE | MSE | MAE | MSE | MAE | MSE | MAE | MSE | MAE | MSE | MAE | MSE | MAE | MSE | MAE |
| ETTh1 | 96 | 0.383 | 0.401 | **0.373** | **0.396** | 0.375 | 0.395 | **0.368** | **0.390** | 0.414 | 0.419 | **0.373** | **0.398** | 0.386 | 0.405 | **0.373** | **0.401** |
| | 192 | 0.437 | 0.432 | **0.435** | 0.434 | 0.436 | 0.428 | **0.424** | **0.424** | 0.460 | 0.445 | **0.424** | **0.427** | 0.441 | 0.436 | **0.432** | **0.436** |
| | 336 | 0.494 | 0.461 | **0.474** | **0.442** | 0.496 | 0.455 | **0.470** | **0.440** | 0.501 | 0.466 | **0.465** | **0.447** | 0.487 | 0.458 | **0.466** | **0.455** |
| | 720 | 0.540 | 0.499 | **0.464** | **0.459** | 0.520 | 0.484 | **0.462** | **0.461** | 0.500 | 0.488 | **0.444** | **0.458** | 0.503 | 0.491 | **0.455** | **0.466** |
| | Avg. | 0.464 | 0.448 | **0.437** | **0.433** | 0.457 | 0.441 | **0.431** | **0.429** | 0.469 | 0.455 | **0.427** | **0.433** | 0.454 | 0.448 | **0.431** | **0.440** |
| ETTh2 | 96 | 0.299 | 0.345 | **0.283** | **0.336** | 0.298 | 0.344 | **0.280** | **0.333** | 0.302 | 0.348 | **0.285** | **0.336** | 0.297 | 0.349 | **0.293** | **0.342** |
| | 192 | 0.371 | 0.394 | **0.362** | **0.385** | 0.372 | 0.396 | **0.357** | **0.384** | 0.388 | 0.400 | **0.367** | **0.389** | 0.380 | 0.400 | **0.373** | **0.393** |
| | 336 | 0.420 | 0.429 | **0.404** | **0.420** | 0.431 | 0.439 | **0.400** | **0.420** | 0.426 | 0.433 | **0.409** | **0.425** | 0.428 | 0.432 | **0.417** | **0.429** |
| | 720 | 0.438 | 0.450 | **0.413** | **0.435** | 0.450 | 0.458 | **0.409** | **0.436** | 0.431 | 0.446 | **0.419** | **0.442** | 0.427 | 0.445 | **0.424** | **0.442** |
| | Avg. | 0.382 | 0.405 | **0.366** | **0.394** | 0.388 | 0.409 | **0.362** | **0.393** | 0.387 | 0.407 | **0.370** | **0.398** | 0.383 | 0.407 | **0.377** | **0.402** |
| ETTm1 | 96 | 0.327 | 0.366 | **0.325** | **0.361** | 0.319 | 0.360 | **0.306** | **0.349** | 0.329 | 0.367 | **0.316** | **0.354** | 0.334 | 0.368 | **0.315** | **0.353** |
| | 192 | 0.370 | 0.386 | **0.367** | **0.383** | 0.360 | 0.381 | **0.349** | **0.375** | 0.367 | 0.385 | **0.360** | **0.382** | 0.377 | 0.391 | **0.369** | **0.387** |
| | 336 | 0.404 | 0.410 | **0.400** | **0.405** | 0.389 | 0.403 | **0.379** | **0.394** | 0.399 | 0.410 | **0.393** | **0.402** | 0.426 | 0.420 | **0.403** | **0.412** |
| | 720 | 0.462 | 0.445 | **0.462** | **0.443** | 0.447 | 0.441 | **0.438** | **0.435** | 0.454 | 0.439 | **0.454** | **0.437** | 0.491 | 0.459 | **0.460** | **0.445** |
| | Avg. | 0.391 | 0.402 | **0.388** | **0.398** | 0.379 | 0.396 | **0.368** | **0.388** | 0.387 | 0.400 | **0.381** | **0.394** | 0.407 | 0.410 | **0.387** | **0.399** |
| ETTm2 | 96 | 0.178 | 0.259 | **0.175** | **0.259** | 0.163 | 0.246 | **0.161** | **0.244** | 0.175 | 0.259 | **0.176** | **0.261** | 0.180 | 0.264 | **0.179** | **0.264** |
| | 192 | 0.242 | 0.302 | **0.240** | **0.300** | 0.229 | 0.290 | **0.225** | **0.286** | 0.241 | 0.302 | **0.241** | **0.300** | 0.250 | 0.309 | **0.241** | **0.302** |
| | 336 | 0.299 | 0.340 | **0.295** | **0.336** | 0.284 | 0.437 | **0.282** | **0.323** | 0.305 | 0.343 | **0.303** | **0.340** | 0.311 | 0.348 | **0.305** | **0.343** |
| | 720 | 0.400 | 0.398 | **0.394** | **0.394** | 0.389 | 0.391 | **0.380** | **0.384** | 0.402 | 0.400 | **0.404** | **0.403** | 0.412 | 0.407 | **0.406** | **0.400** |
| | Avg. | 0.280 | 0.325 | **0.276** | **0.322** | 0.266 | 0.341 | **0.262** | **0.309** | 0.281 | 0.326 | **0.281** | **0.326** | 0.288 | 0.332 | **0.283** | **0.327** |
| Electricity | 96 | 0.182 | 0.265 | **0.181** | **0.264** | 0.136 | 0.229 | **0.134** | **0.228** | 0.181 | 0.270 | **0.163** | **0.248** | 0.148 | 0.240 | **0.145** | **0.237** |
| | 192 | 0.187 | 0.270 | **0.186** | **0.268** | 0.152 | 0.244 | **0.152** | **0.244** | 0.188 | 0.274 | **0.172** | **0.257** | 0.162 | 0.253 | **0.158** | **0.252** |
| | 336 | 0.203 | 0.287 | **0.202** | **0.285** | 0.170 | 0.264 | **0.170** | **0.263** | 0.204 | 0.293 | **0.191** | **0.278** | 0.178 | 0.269 | **0.176** | **0.271** |
| | 720 | 0.244 | 0.319 | **0.243** | **0.317** | 0.212 | 0.299 | **0.210** | **0.296** | 0.246 | 0.324 | **0.239** | **0.319** | 0.225 | 0.317 | **0.212** | **0.306** |
| | Avg. | 0.204 | 0.285 | **0.203** | **0.283** | 0.168 | 0.259 | **0.166** | **0.258** | 0.205 | 0.290 | **0.191** | **0.275** | 0.178 | 0.270 | **0.173** | **0.267** |
| Traffic | 96 | 0.510 | 0.331 | **0.477** | **0.301** | 0.458 | 0.296 | **0.420** | **0.275** | 0.462 | 0.295 | **0.433** | **0.280** | 0.395 | 0.268 | **0.374** | **0.247** |
| | 192 | 0.505 | 0.327 | **0.478** | **0.300** | 0.457 | 0.295 | **0.437** | **0.283** | 0.466 | 0.296 | **0.447** | **0.287** | 0.417 | 0.276 | **0.399** | **0.259** |
| | 336 | 0.518 | 0.332 | **0.492** | **0.305** | 0.470 | 0.299 | **0.453** | **0.291** | 0.482 | 0.304 | **0.455** | **0.285** | 0.433 | 0.283 | **0.419** | **0.269** |
| | 720 | 0.553 | 0.350 | **0.531** | **0.328** | 0.502 | 0.314 | **0.482** | **0.310** | 0.514 | 0.322 | **0.486** | **0.302** | 0.467 | 0.302 | **0.451** | **0.291** |
| | Avg. | 0.522 | 0.335 | **0.494** | **0.308** | 0.472 | 0.301 | **0.448** | **0.290** | 0.481 | 0.304 | **0.455** | **0.288** | 0.428 | 0.282 | **0.411** | **0.266** |
| Weather | 96 | 0.181 | 0.219 | **0.176** | **0.214** | 0.158 | 0.203 | **0.158** | **0.203** | 0.177 | 0.218 | **0.169** | **0.211** | 0.174 | 0.214 | **0.173** | **0.213** |
| | 192 | 0.228 | 0.259 | **0.223** | **0.256** | 0.207 | 0.247 | **0.206** | **0.244** | 0.225 | 0.259 | **0.215** | **0.251** | 0.221 | 0.254 | **0.223** | **0.257** |
| | 336 | 0.282 | 0.299 | **0.279** | **0.295** | 0.262 | 0.289 | **0.260** | **0.285** | 0.278 | 0.297 | **0.274** | **0.293** | 0.278 | 0.296 | **0.278** | **0.296** |
| | 720 | 0.357 | 0.347 | **0.355** | **0.345** | 0.344 | 0.344 | **0.343** | **0.342** | 0.354 | 0.348 | **0.353** | **0.345** | 0.358 | 0.347 | **0.353** | **0.344** |
| | Avg. | 0.262 | 0.281 | **0.258** | **0.278** | 0.243 | 0.271 | **0.242** | **0.268** | 0.259 | 0.281 | **0.253** | **0.275** | 0.258 | 0.278 | **0.257** | **0.278** |
| Imp% | Avg. | - | - | 3.24% | 2.60% | - | - | 3.92% | 3.40% | - | - | 4.48% | 2.97% | - | - | 3.24% | 1.94% |

Table 9: Full results for performance comparisons between backbone models and them integrated with our proposals on ETT, electricity, traffic and weather datasets. All baseline results are run ourselves

| Models | | MLP | | +Ours | | CYCLENET | | +Ours | | PATCHTST | | +Ours | | ITRANS | | +Ours | |
|---|---|---|---|---|---|---|---|---|---|---|---|---|---|---|---|---|---|
| Metric | | MSE | MAE | MSE | MAE | MSE | MAE | MSE | MAE | MSE | MAE | MSE | MAE | MSE | MAE | MSE | MAE |
| ETTh1 | 96 | 0.380 | 0.399 | 0.373 | 0.396 | 0.379 | 0.400 | 0.368 | 0.390 | 0.390 | 0.407 | 0.373 | 0.398 | 0.383 | 0.405 | 0.373 | 0.401 |
| | 192 | 0.444 | 0.428 | 0.435 | 0.434 | 0.437 | 0.432 | 0.424 | 0.424 | 0.461 | 0.446 | 0.424 | 0.427 | 0.434 | 0.433 | 0.432 | 0.436 |
| | 336 | 0.478 | 0.444 | 0.474 | 0.442 | 0.477 | 0.446 | 0.470 | 0.440 | 0.486 | 0.457 | 0.465 | 0.447 | 0.470 | 0.452 | 0.466 | 0.455 |
| | 720 | 0.478 | 0.475 | 0.464 | 0.459 | 0.469 | 0.466 | 0.462 | 0.461 | 0.484 | 0.469 | 0.444 | 0.458 | 0.464 | 0.470 | 0.455 | 0.466 |
| | Avg. | 0.445 | 0.437 | 0.437 | 0.433 | 0.441 | 0.436 | 0.431 | 0.429 | 0.455 | 0.445 | 0.427 | 0.433 | 0.438 | 0.440 | 0.431 | 0.440 |
| ETTh2 | 96 | 0.293 | 0.343 | 0.283 | 0.336 | 0.297 | 0.347 | 0.280 | 0.333 | 0.298 | 0.345 | 0.285 | 0.336 | 0.321 | 0.362 | 0.293 | 0.342 |
| | 192 | 0.368 | 0.391 | 0.362 | 0.385 | 0.374 | 0.396 | 0.357 | 0.384 | 0.394 | 0.401 | 0.367 | 0.389 | 0.394 | 0.408 | 0.373 | 0.393 |
| | 336 | 0.419 | 0.427 | 0.404 | 0.420 | 0.417 | 0.432 | 0.400 | 0.420 | 0.418 | 0.429 | 0.409 | 0.425 | 0.449 | 0.447 | 0.417 | 0.429 |
| | 720 | 0.427 | 0.443 | 0.413 | 0.435 | 0.430 | 0.447 | 0.409 | 0.436 | 0.437 | 0.454 | 0.419 | 0.442 | 0.435 | 0.449 | 0.424 | 0.442 |
| | Avg. | 0.377 | 0.401 | 0.366 | 0.394 | 0.380 | 0.406 | 0.362 | 0.393 | 0.387 | 0.407 | 0.370 | 0.398 | 0.400 | 0.417 | 0.377 | 0.402 |
| ETTm1 | 96 | 0.348 | 0.371 | 0.325 | 0.361 | 0.315 | 0.358 | 0.306 | 0.349 | 0.339 | 0.370 | 0.316 | 0.354 | 0.351 | 0.378 | 0.315 | 0.353 |
| | 192 | 0.388 | 0.391 | 0.367 | 0.383 | 0.359 | 0.382 | 0.349 | 0.375 | 0.381 | 0.393 | 0.360 | 0.382 | 0.393 | 0.399 | 0.369 | 0.387 |
| | 336 | 0.422 | 0.412 | 0.400 | 0.405 | 0.389 | 0.407 | 0.379 | 0.394 | 0.411 | 0.413 | 0.393 | 0.402 | 0.422 | 0.421 | 0.403 | 0.412 |
| | 720 | 0.493 | 0.451 | 0.462 | 0.443 | 0.454 | 0.441 | 0.438 | 0.435 | 0.474 | 0.449 | 0.454 | 0.437 | 0.487 | 0.460 | 0.460 | 0.445 |
| | Avg. | 0.413 | 0.406 | 0.388 | 0.398 | 0.379 | 0.397 | 0.368 | 0.388 | 0.401 | 0.406 | 0.381 | 0.394 | 0.413 | 0.414 | 0.387 | 0.399 |
| ETTm2 | 96 | 0.186 | 0.270 | 0.175 | 0.259 | 0.164 | 0.248 | 0.161 | 0.244 | 0.180 | 0.263 | 0.176 | 0.261 | 0.189 | 0.275 | 0.179 | 0.264 |
| | 192 | 0.249 | 0.309 | 0.240 | 0.300 | 0.228 | 0.289 | 0.225 | 0.286 | 0.248 | 0.310 | 0.241 | 0.300 | 0.260 | 0.318 | 0.241 | 0.302 |
| | 336 | 0.308 | 0.345 | 0.295 | 0.336 | 0.285 | 0.328 | 0.282 | 0.323 | 0.307 | 0.345 | 0.303 | 0.340 | 0.326 | 0.359 | 0.305 | 0.343 |
| | 720 | 0.404 | 0.398 | 0.394 | 0.394 | 0.387 | 0.387 | 0.380 | 0.384 | 0.411 | 0.404 | 0.404 | 0.403 | 0.423 | 0.412 | 0.406 | 0.400 |
| | Avg. | 0.287 | 0.331 | 0.276 | 0.322 | 0.266 | 0.313 | 0.262 | 0.309 | 0.287 | 0.331 | 0.281 | 0.326 | 0.299 | 0.341 | 0.283 | 0.327 |
| Electricity | 96 | 0.187 | 0.267 | 0.181 | 0.264 | 0.136 | 0.230 | 0.134 | 0.228 | 0.181 | 0.268 | 0.163 | 0.248 | 0.161 | 0.251 | 0.145 | 0.237 |
| | 192 | 0.191 | 0.272 | 0.186 | 0.268 | 0.154 | 0.246 | 0.152 | 0.244 | 0.193 | 0.277 | 0.172 | 0.257 | 0.178 | 0.267 | 0.158 | 0.252 |
| | 336 | 0.206 | 0.288 | 0.202 | 0.285 | 0.171 | 0.264 | 0.170 | 0.263 | 0.199 | 0.286 | 0.191 | 0.278 | 0.193 | 0.281 | 0.176 | 0.271 |
| | 720 | 0.251 | 0.325 | 0.243 | 0.317 | 0.212 | 0.299 | 0.210 | 0.296 | 0.240 | 0.319 | 0.239 | 0.319 | 0.213 | 0.306 | 0.212 | 0.306 |
| | Avg. | 0.209 | 0.288 | 0.203 | 0.283 | 0.168 | 0.260 | 0.166 | 0.258 | 0.203 | 0.287 | 0.191 | 0.275 | 0.186 | 0.276 | 0.173 | 0.267 |
| Traffic | 96 | 0.504 | 0.314 | 0.477 | 0.301 | 0.432 | 0.292 | 0.420 | 0.275 | 0.468 | 0.301 | 0.433 | 0.280 | 0.403 | 0.276 | 0.374 | 0.247 |
| | 192 | 0.522 | 0.330 | 0.478 | 0.300 | 0.443 | 0.294 | 0.437 | 0.283 | 0.474 | 0.305 | 0.447 | 0.287 | 0.423 | 0.283 | 0.399 | 0.259 |
| | 336 | 0.534 | 0.334 | 0.492 | 0.305 | 0.460 | 0.304 | 0.453 | 0.291 | 0.498 | 0.324 | 0.455 | 0.285 | 0.441 | 0.292 | 0.419 | 0.269 |
| | 720 | 0.540 | 0.337 | 0.531 | 0.328 | 0.489 | 0.321 | 0.482 | 0.310 | 0.547 | 0.340 | 0.486 | 0.302 | 0.465 | 0.303 | 0.451 | 0.291 |
| | Avg. | 0.525 | 0.329 | 0.494 | 0.308 | 0.456 | 0.303 | 0.448 | 0.290 | 0.497 | 0.317 | 0.455 | 0.288 | 0.433 | 0.288 | 0.411 | 0.266 |
| Weather | 96 | 0.192 | 0.232 | 0.176 | 0.214 | 0.165 | 0.209 | 0.158 | 0.203 | 0.185 | 0.225 | 0.169 | 0.211 | 0.211 | 0.257 | 0.173 | 0.213 |
| | 192 | 0.237 | 0.266 | 0.223 | 0.256 | 0.209 | 0.248 | 0.206 | 0.244 | 0.229 | 0.262 | 0.215 | 0.251 | 0.255 | 0.285 | 0.223 | 0.257 |
| | 336 | 0.290 | 0.304 | 0.279 | 0.295 | 0.269 | 0.292 | 0.260 | 0.285 | 0.283 | 0.302 | 0.274 | 0.293 | 0.303 | 0.319 | 0.278 | 0.296 |
| | 720 | 0.366 | 0.352 | 0.355 | 0.345 | 0.348 | 0.344 | 0.343 | 0.342 | 0.361 | 0.351 | 0.353 | 0.345 | 0.370 | 0.362 | 0.353 | 0.344 |
| | Avg. | 0.271 | 0.288 | 0.258 | 0.278 | 0.248 | 0.273 | 0.242 | 0.268 | 0.264 | 0.285 | 0.253 | 0.275 | 0.285 | 0.306 | 0.257 | 0.278 |
| Imp% | Avg. | - | - | 4.11% | 2.56% | - | - | 2.52% | 2.18% | - | - | 5.48% | 3.61% | - | - | 5.51% | 4.17% |

Table 10: Full results of ablation study with backbone predictor as CYCLENET.

| Models | | CYCLENET | | +SNR | | +SCAM | | +both | |
|---|---|---|---|---|---|---|---|---|---|
| Metric | | MSE | MAE | MSE | MAE | MSE | MAE | MSE | MAE |
| ETTh1 | 96 | 0.381 | 0.399 | 0.374 | 0.392 | 0.372 | 0.393 | 0.368 | 0.390 |
| | 192 | 0.438 | 0.426 | 0.432 | 0.423 | 0.424 | 0.424 | 0.424 | 0.424 |
| | 336 | 0.478 | 0.446 | 0.475 | 0.444 | 0.470 | 0.440 | 0.470 | 0.440 |
| | 720 | 0.478 | 0.472 | 0.470 | 0.469 | 0.480 | 0.473 | 0.462 | 0.461 |
| | Avg. | 0.444 | 0.436 | 0.438 | 0.432 | 0.436 | 0.432 | 0.431 | 0.429 |
| ETTh2 | 96 | 0.297 | 0.347 | 0.288 | 0.339 | 0.283 | 0.336 | 0.280 | 0.333 |
| | 192 | 0.379 | 0.399 | 0.371 | 0.392 | 0.357 | 0.384 | 0.357 | 0.384 |
| | 336 | 0.419 | 0.433 | 0.412 | 0.425 | 0.400 | 0.420 | 0.400 | 0.420 |
| | 720 | 0.430 | 0.447 | 0.415 | 0.437 | 0.421 | 0.440 | 0.409 | 0.436 |
| | Avg. | 0.381 | 0.407 | 0.372 | 0.398 | 0.365 | 0.395 | 0.362 | 0.393 |
| ETTm1 | 96 | 0.315 | 0.358 | 0.312 | 0.353 | 0.308 | 0.349 | 0.306 | 0.349 |
| | 192 | 0.359 | 0.382 | 0.356 | 0.379 | 0.352 | 0.372 | 0.349 | 0.375 |
| | 336 | 0.389 | 0.407 | 0.384 | 0.400 | 0.385 | 0.402 | 0.379 | 0.394 |
| | 720 | 0.454 | 0.441 | 0.449 | 0.435 | 0.441 | 0.436 | 0.438 | 0.435 |
| | Avg. | 0.379 | 0.397 | 0.375 | 0.392 | 0.371 | 0.390 | 0.368 | 0.388 |
| ETTm2 | 96 | 0.164 | 0.248 | 0.162 | 0.245 | 0.161 | 0.244 | 0.161 | 0.244 |
| | 192 | 0.228 | 0.289 | 0.227 | 0.287 | 0.226 | 0.287 | 0.225 | 0.286 |
| | 336 | 0.285 | 0.328 | 0.285 | 0.326 | 0.283 | 0.326 | 0.282 | 0.323 |
| | 720 | 0.387 | 0.387 | 0.385 | 0.385 | 0.381 | 0.386 | 0.380 | 0.384 |
| | Avg. | 0.266 | 0.313 | 0.265 | 0.311 | 0.263 | 0.311 | 0.262 | 0.309 |
| Weather | 96 | 0.165 | 0.209 | 0.159 | 0.203 | 0.159 | 0.202 | 0.158 | 0.203 |
| | 192 | 0.209 | 0.248 | 0.210 | 0.249 | 0.206 | 0.244 | 0.206 | 0.244 |
| | 336 | 0.269 | 0.292 | 0.267 | 0.291 | 0.260 | 0.285 | 0.260 | 0.285 |
| | 720 | 0.348 | 0.344 | 0.347 | 0.343 | 0.343 | 0.338 | 0.343 | 0.342 |
| | Avg. | 0.248 | 0.273 | 0.246 | 0.272 | 0.242 | 0.267 | 0.242 | 0.268 |

Table 11: Full results of ablation study with backbone predictor as ɪTRANS.

| Models | | ɪTRANS | | +SNR | | +SCAM | | +both | |
|--------|------|-------|-------|-------|-------|-------|-------|-------|-------|
| Metric | | MSE | MAE | MSE | MAE | MSE | MAE | MSE | MAE |
| ETTh1 | 96 | 0.383 | 0.405 | 0.388 | 0.407 | 0.376 | 0.397 | **0.373** | **0.401** |
| | 192 | 0.434 | 0.433 | 0.446 | 0.441 | 0.434 | 0.436 | **0.432** | **0.436** |
| | 336 | 0.470 | 0.452 | 0.472 | 0.457 | 0.470 | 0.451 | **0.466** | **0.455** |
| | 720 | 0.464 | 0.470 | 0.467 | 0.475 | 0.464 | 0.467 | **0.455** | **0.466** |
| | Avg. | 0.438 | 0.440 | 0.443 | 0.445 | 0.436 | 0.438 | **0.431** | **0.440** |
| ETTh2 | 96 | 0.321 | 0.362 | 0.331 | 0.370 | 0.296 | 0.344 | **0.293** | **0.342** |
| | 192 | 0.394 | 0.408 | 0.398 | 0.410 | 0.385 | 0.399 | **0.373** | **0.393** |
| | 336 | 0.449 | 0.447 | 0.431 | 0.437 | 0.418 | 0.429 | **0.417** | **0.429** |
| | 720 | 0.435 | 0.449 | 0.437 | 0.453 | 0.426 | 0.443 | **0.424** | **0.442** |
| | Avg. | 0.400 | 0.417 | 0.399 | 0.417 | 0.381 | 0.404 | **0.377** | **0.402** |
| ETTm1 | 96 | 0.351 | 0.378 | 0.345 | 0.376 | 0.324 | 0.366 | **0.315** | **0.353** |
| | 192 | 0.393 | 0.399 | 0.385 | 0.395 | 0.376 | 0.393 | **0.369** | **0.387** |
| | 336 | 0.422 | 0.421 | 0.429 | 0.427 | 0.404 | 0.412 | **0.403** | **0.412** |
| | 720 | 0.487 | 0.460 | 0.494 | 0.461 | 0.461 | 0.447 | **0.460** | **0.445** |
| | Avg. | 0.413 | 0.414 | 0.413 | 0.415 | 0.391 | 0.404 | **0.387** | **0.399** |
| ETTm2 | 96 | 0.189 | 0.275 | 0.204 | 0.289 | 0.184 | 0.279 | **0.179** | **0.264** |
| | 192 | 0.260 | 0.318 | 0.275 | 0.331 | 0.252 | 0.321 | **0.241** | **0.302** |
| | 336 | 0.326 | 0.359 | 0.324 | 0.358 | 0.323 | 0.357 | **0.305** | **0.343** |
| | 720 | 0.423 | 0.412 | 0.421 | 0.410 | 0.412 | 0.411 | **0.406** | **0.400** |
| | Avg. | 0.299 | 0.341 | 0.306 | 0.347 | 0.293 | 0.342 | **0.283** | **0.327** |
| Weather | 96 | 0.211 | 0.257 | 0.212 | 0.247 | 0.190 | 0.226 | **0.173** | **0.213** |
| | 192 | 0.255 | 0.285 | 0.252 | 0.283 | 0.243 | 0.270 | **0.223** | **0.257** |
| | 336 | 0.303 | 0.319 | 0.301 | 0.319 | 0.295 | 0.307 | **0.278** | **0.296** |
| | 720 | 0.370 | 0.362 | 0.367 | 0.357 | 0.366 | 0.353 | **0.353** | **0.344** |
| | Avg. | 0.285 | 0.306 | 0.283 | 0.302 | 0.274 | 0.289 | **0.257** | **0.278** |

