# OpenReview forum: "Not All Data are Good Labels: On the Self-supervised Labeling for Time Series Forecasting"
_NeurIPS.cc/2025/Conference — NeurIPS 2025 spotlight_

### Official Review · Reviewer_GL6M · 2025-06-23

**Clarity:** 4
**Significance:** 4
**Originality:** 4
**Rating:** 5
**Confidence:** 4

**Summary:**

This paper focuses on data quality for enhancing TSF by proposing a self-supervised paradigm. The core insight is that raw time-series data, inherently as labels, often contain noise or anomalies, which degrade model performance. To mitigate this, the authors introduce ​​Self-Correction with Adaptive Mask (SCAM)​​, which generates pseudo-labels via an auxiliary reconstruction network. SCAM identifies and replaces overfitted label components with reconstructed pseudo-labels, while ​​Spectral Norm Regularization (SNR)​​ suppresses overfitting in this procedure. Experiments across representative datasets and diverse backbones (e.g., MLP, Transformer-based models) show consistent improvements in generalization. The method decouples data quality assessment from prediction, offering a novel angle for TSF robustness.

**Questions:**

Q1. How are SNR-normalized parameters initialized? Are original weights retained and spectrally normalized during training, or is SNR applied as a reparameterization?

Q2. Could you explicitly define the dimensions of "quality" used in this work? For example: 1) Is "label quality" solely measured by noise magnitude (e.g., SNR), or does it include temporal consistency/plausibility? 2) For reconstructed pseudo-labels (Sec 3.2), is quality assessed via ℓ_rec (reconstruction error) or downstream ℓ_target (forecasting accuracy)?

**Ethical Concerns:**

["NO or VERY MINOR ethics concerns only"]

**Final Justification:**

The author's rebuttal addressed all of my concerns. I will maintain my current positive score and support its acceptance.

**Limitations:**

Yes

**Quality:**

4

**Strengths And Weaknesses:**

Strengths:

S1. Telling a story with intuitive visualizations and the theory is clearly derived, e.g., Eq 4 for loss decomposition and Fig 6 for adaptive masked parts.

S2. Addressing understudied label quality issues in TSF is of interest. The self-supervised paradigm reduces dependency on curated data, benefiting real-world applications --- important for practical scenarios where data are noisy without known ground truth.

S3. Novel formulation of label noise via reconstruction-based candidate datasets. SCAM's adaptive masking and SNR integration are inventive with insights into time series data and current strong models.

S4. Experimentally validated across diverse settings; Appendix details aid replication; particularly, the reproductivity report in G.3 and efficiency study in G.4 are helpful.

Weaknesses:

W1. SNR implementation details (Sec 3.4) can be clearer; a brief pseudocode would help.

W2. The "quality" discussed in the paper should be made explicit.

W3. Some presentation issues, like Fig 14 on Page 18 and widow in line 310 (should be minor).

---

> ### Author Rebuttal · Authors · 2025-07-31
>
> **Q1: implementation details of SNR**
>
> **(1) Power iteration to compute spectral norm**
> SNR is implemented in a reparameterization manner, meaning that the initialization of SNR-applied parameters follows the conventional Xavier initialization.
>
> The SNR ensures a parameter matrix $W$ has a spectral norm $\Vert W\Vert_{2}$ exactly equals to 1. Essentially what SNR substitutes $W$ with a normalized matrix $\frac{W}{||W||_{2}}$.
>
> To normalize matrix $W$, we first calculate its the spectral norm $\Vert W\Vert_{2}$, which is  defined as $$\Vert W\Vert_{2} =\max \frac{\Vert Wx\Vert_{2}}{\Vert x\Vert_{2}}=\max \frac{x^{\top}W^{\top}Wx}{x^\top x}.$$This corresponds to the largest singular value of $W^{\top}W$.
>
> We compute this value numerically using **power iteration**, applying the following update rule iteratively:
>
> $$u_{i+1}=\frac{W^{\top}Wu_i}{||W^{\top}Wu_i||}$$
>
> After several iterations, we obtain $||W||^2_2 \approx u_t^{\top}W^{\top}Wu_t$. The initial value of $u$ can be set as all-ones vector.
>
> **(2) Brief Proof of Convergence**
>
> Let the singular values of $W^{\top}W$ be $\lambda_1 > \lambda_2 > \dots >\lambda_k$, with corresponding singular vectors $\eta_1, \eta_2, \dots, \eta_k$ forming a basis of a k-dimension vector space.
> Expressing $u_i$ in this basis, we have:
> $$u_i = c_1 \eta_1 + c_2\eta_2 + ... + c_k\eta_k$$The iteration becomes:
> $$u_{i+1} = c_1\lambda_1\eta_1 + c_2\lambda_2\eta_2 + ... + c_k\lambda_k\eta_k$$
> Let $l_{i}$ be $||W^{\top}Wu_i||$. After $t$ iterations, we have: $$u_t / \lambda_1^t = (\prod c_1 l_i) \eta_1 + (\prod c_2 l_i) (\frac{\lambda_2}{\lambda_1})^t + ... + (\prod c_k l_i) (\frac{\lambda_k}{\lambda_1})^t.$$As $t \to \infty$, the terms $(\frac{\lambda i}{\lambda_1})^t\to 0$, leaving $$u_t \approx \lambda_1^t (\prod c_1l_i)\eta_1.$$Due to  normalization in each iteration, $u_t$ converges to a unit vector aligned with $\eta_1$, yielding:   $$||W||_2^2=\eta_1^{\top}W^{\top}W\eta_1 \approx u_t^{\top}W^{\top}Wu_t.$$
>
> **(3) Implementation details**
> In practice, we stop gradients (via `tensor.detach()`) for both $W$ and $u_i$​ during power iteration. This means the original parameter $W$ in a linear layer is scaled only by a constant. The full codes regarding SNR are available in our code base (in `src/models/modules/snrlinear.py`)
>
> We are happy to provide pseudocode and supplementary illustrations in the camera-ready version to further clarify the SNR method.
>
> **Q2: Definition of Label Quality**
>
> **(1) Current Metrics for Label Quality**
>
> Label quality is evaluated using end-to-end test metrics. Given a raw dataset $A$, we split it into training set $A_{train}$ and test set $A_{test}$. Our reconstruction method generates a corresponding dataset $B_{train}$ that is exclusively used for training. We then train two separate predictors: $f_A$ on the original dataset $A$ and $f_B$ on the reconstructed dataset $B$. When comparing their performance on $A_{test}$, if $f_{B}$ demonstrates lower prediction error (i.e., better performance), we conclude that the labels in $B_{train}$ are of higher quality. Thus, the quality of our reconstructed pseudo-labels is ultimately assessed through the downstream target metric $\ell_{target}$.
>
> **(2) More Information-Rich Metrics**
>
> We propose that sharpness ($\lambda_{max}$) [1], as shown in Figure 7, serves as an effective metric for evaluating a model's robustness to data noise. This metric, combined with the SNR score of parameter matrices, focuses more on model characteristics rather than direct data properties. We fully agree that temporal consistency and plausibility should be considered when assessing data noise. However, current techniques still lack methods that can directly leverage prior knowledge about noise patterns in the data. We greatly appreciate your suggestion and look forward to expanding the self-supervised learning framework toward more model-free and data-centric approaches.
>
> _[1] Samformer: Unlocking the potential of transformers in time series forecasting with sharpness-aware minimization and channel-wise attention._

---

> > ### Comment · Reviewer_GL6M · 2025-08-05
> >
> > The author's rebuttal addressed all of my concerns. I will maintain my current positive score and support its acceptance.

---

### Official Review · Reviewer_wXrM · 2025-06-25

**Clarity:** 2
**Significance:** 3
**Originality:** 3
**Rating:** 5
**Confidence:** 3

**Summary:**

This study devised a new data augmentation technique, using a custom adaptive mask and a CNN network, for time series data which create pseudo-labels over the time-series, smoothing the loss landscape and enhancing the generalization of the predictor model.The technique created, the Self-Correction with Adaptive Mask (SCAM) and the use of Spectral Norm Regularization (SNR) were tested on real-world datasets by additioning their procedures to known techniques such as MLP, CycleNet, PatchTST and ITransformer. Of note, their procedure discards overfitted components, and is applicable when one wants to work on the prediction of each element in a time-series, while others work on a set of elements.

**Questions:**

1. How many replicates have been performed for the different experiments?
2. Do the method(s) apply to both short-term and long-term forecasting?
3. How does the N (number of candidates) and J (steps) determined - Algorithm 1?
4. Are there any cases when the reconstructed networks diverge too much from the true labels and cause more error than leaving the original data? In figure 8, it seems like for low noise, it actually makes the prediction less accurate?

**Ethical Concerns:**

["NO or VERY MINOR ethics concerns only"]

**Final Justification:**

Dear editors,

The authors answered all my questions. The answers to the other reviewers also greatly improved the comprehension of the paper and the results.

Looking forward to a revised version of the paper, including the better description of Algo. 1. to be included in the paper. Still, I think some of the text should be revised in  the final version to add more context to this work, and rely less on the Annexe section.

I adjusted the status of the paper from 'Borderline reject' to 'Accepted' following the authors answers.

**Limitations:**

No societal impact was discussed in the paper.

One possible avenue would be to discuss the implication for wather forecasting, traffic or electricity distribution since the authors used datasets in those specific domains. For example, better forecasting in agriculture?

**Paper Formatting Concerns:**

The figures (e.g. 3-8) in the main paper are too small to be comprehensive.

**Quality:**

2

**Strengths And Weaknesses:**

*Strengths
- Innovative technique to mask some part of the time-series so the deep learning network can leverage the “best points” when training, facilitating the training process.
- Results seems to indicate some improvement over the existing SOTA TSF methods.
- Good definitions for the different loss functions and the objectives are well defined.

*Weaknesses

- The authors introduced two questions in the introduction which are never answered in the study (line 26 and 27).
- New concepts, important for the method (e.g. Conv-concat Layer - Figure 2?), are only discussed in the Annexe, but are central to the reconstruction networks.
- Related-work could have been longer with more contextual information on concurrent methods and the tested deep learning network (e.g. ITransformer, CycleNet). The information is available in the paper, e.g. lines 276-280, but it's out of place and limited. For example, annexe is far more informative and section D.2 should be put in the main section.
- Figures and tables are too small to be informative in the main paper. No formal statistics for the results and no information on the number of replicates. Furthermore, the difference between with/without addition of SCAM on the results is small.
- Source code on the git could have been simplified so that only the relevant code (or some sample notebook) be available.
- Experimental procedure for the results provided (Table 1) are unclear and sould be provided in the main text.

---

> ### Author Rebuttal · Authors · 2025-07-31
>
> **About weaknesses**
>
> **w1 two questions in the introduction:** We answer the questions right next to them (line 29, "We posit that both answers are positive..."). Admittedly, the answers are not directly given with concise reasons. The whole idea of our paper is to construct pseudo labels natively from raw datasets that increase the quality of labels. We have backed this claim throughout the paper.
>
> **w2 introduction of the reconstruction network:** We have a detailed introduction of our implementation in Appendix D. The idea to adaptively provide pseudo labels during the training of a TSF model is novel and therefore leaves room for more extensive discussion on the designs of the reconstruction network. We simply provide one choice and do not claim its optimality or SOTA performance on this novel task.
>
> **w3 about related work:** We will make adjustments accordingly, and do our best to satisfy NeurIPS's formatting requirements.
>
> **w4 presentation of figures and tables/main experiments results:**
>
> Regarding the figures and tables, we are happy to provide a more reader-friendly version after releasing our code repository (Sorry that we cannot provide them right now due to NeurIPS's rebuttal policy).
>
> For statistical results of the main experiments, we conduct 1 replicate only for all experiments (under a fixed random seed). The main experiments (Table 7, Table 8 in the paper) contain 352 individual experiments, each of which approximately costs 0.6 to 8 GPU hours, adding up to around 1000 GPU hours.
>
> The statistical significance is guaranteed by extensive experiments already. Our method shows improved performance against 4 different baselines on 11 datasets, 8 different forecasting horizons.
>
> Regarding reproductivity, we have provided detailed guides in Appendix G. Additionally, we ensure our code base can produce exactly the same results when fixing the random seed (we use 1234 as the seed in all experiments).
>
> **w5 about** **source code** **and simple demos:** We are happy to provide a more illustrative notebook and detailed documents upon acceptance (we cannot revise the code base due to NeurIPS policy). For the current code repository, we provide the full framework, which enables free combination of different model components by simple revisions of the config files. Additionally, the pytorch lightning framework that we have adopted works cleanly with logging tools like wandb and aim, which enable easy management of experiment results. The complete training and evaluation framework is built on our own which we believe will benefit the community.
>
> **w6 experiment procedure in the main text:** We are pleased to consider the arrangement of the current main text content. Please understand that NeurIPS has formatting requirements that only allow for a very concise main text.
>
> **Q1 number of replicates for different experiments:**
> We use 1 replicate, which is explained in w4.
>
> **Q2 application to both short-term and long-term forecasting:**
> Yes. Table 7 is the experiment results on short-term forecasting on 4 PeMS datasets, and Table 8 is the experiment results on long-term forecasting on 7 datasets (which is also reported in the main text).
>
> **Q3 Regarding Algorithm 1 in the Initial Case**
>
>  **(1) Background of Algorithm 1**
>
> In Section 3.1 (_Initial Case_), we implement a grid-search-like algorithm that constructs multiple candidate datasets to train a predictor from scratch.
> Let $A = \{a_i\} $ and $B = \{b_i\}$ be two datasets with the same number of samples.
> The distance between them is defined as $d(A, B) = \sum |a_i - b_i|$.
>
> A candidate dataset can be parameterized based on a raw dataset $A$ as $A' = g(A, \phi) = \{g(a_i, \phi)\}$. This allows us to map datasets to an interval $(d_{\text{inf}}, d_{\text{sup}})$, where:
> - $d_{\text{inf}}$ is the lower bound for $d(A, A')$,
> - $d_{\text{sup}}$ is the upper bound.
>
> For any $d_i \in (d_{\text{inf}}, d_{\text{sup}})$, the corresponding $A'_i = g(A, \phi_i)$ is not uniquely determined—there may be infinitely many $\phi_i$ that map to the same $d_i$. To resolve this, we introduce an optimization process as an external constraint.
>
> In each step (Line 11 of Algorithm 1: $\phi_{i+1} \leftarrow \phi_i - \eta \nabla_{\phi}$), we optimize $d(A, A_i')$ (i.e., $\ell_{\text{rec}}$ in Line 11),
> This procedure effectively generates a new dataset $ A_{i+1}' = g(A, \phi_{i+1}) $ that maps to $d_{i+1} \in (d_{\text{inf}}, d_{\text{sup}})$.
> We can then sample as many points in $(d_{\text{inf}}, d_{\text{sup}})$ as desired, with each point corresponding to a unique $\phi_i$ and dataset $A'_i$.
>
>
> We refer to this as a "grid search" because we explore the loss interval via gradient descent, where each point in the interval maps to a distinct dataset. Predictors are then independently trained on these datasets to identify the best-performing one.
>
> **(2) Answers to the Question**
>
> - **$N$ (Number of Optimization Steps):** Combined with the learning rate $\eta$, this hyperparameter determines the lower bound $d_{\text{inf}}$ and the number of samples explored within $(d_{\text{inf}}, d_{\text{sup}})$.
>
> - **$J$ (Termination Condition):** This ensures algorithmic completeness by guaranteeing predictor convergence. We set a gradient threshold $\alpha$ and a maximum optimization step for predictors $f$ as termination criteria.
>
> **(3) Conclusion**
>
> To summarize, Algorithm 1 serves as an illustrative example of the concept, demonstrating that "better" datasets can exist compared to the raw dataset. Here, "better" means a predictor trained on $A'$ achieves superior performance (on $A$'s test set) than one trained on $A$.
>
> We are happy to include these clarifications in the camera-ready version. If further questions arise, we welcome discussion in the following phase.
>
> **Q4: The Fidelity of Reconstructions**
>
> **(1) Explanation of Figure 8**
>
> Figure 8 is provided solely for illustrative purposes, as it depicts results on synthesized data during the early stages of training—when neither the predictor nor the reconstruction network has fully converged. For real-world scenarios, we present actual reconstruction fidelity in Figure 15, which reflects performance during later training phases.
>
> **(2) Empirical Analysis**
>
> To quantitatively assess reconstruction fidelity, we report the following statistics:
>
> **Table 1:** Reconstruction loss ($\ell_{rec}$) after full training (MLP + SCAM). _Horizon_ denotes the forecasting window length (i.e., the time series segment to reconstruct).
>
> ||ETTh1|ETTh2|ETTm1|ETTm2|Weather|Electricity|
> |---|---|---|---|---|---|---|
> |Horizon = 96|2.250E-04|2.435E-04|2.023E-04|2.254E-04|2.012E-04|3.158E-04|
> |Horizon = 336|1.996E-04|2.134E-04|2.055E-04|2.162E-04|1.866E-04|3.281E-04|
>
> The consistently low $\ell_{rec}$ values at convergence confirm that reconstruction fidelity is well-preserved in later training stages. Intuitively:
>
> - **Early stages:** Our method generates easy-to-learn pseudo-labels (which may initially diverge from raw data).
>
> - **Late stages:** Continuous optimization ensures reconstructions converge to recover the original series, enabling predictors to better capture unprocessed temporal patterns.
>
>
> **(3) Architectural Choices to Ensure Fidelity**
>
> Our reconstruction network architecture (see Response to Q2, Reviewer b9qu, for alternatives) provides the strongest theoretical guarantees for fidelity. This is empirically validated by:
>
> 1. **Low $\ell_{rec}$ values**, which correlate with a reduced mask ratio for $M_{<}$ (i.e., greater emphasis on $|\tilde{y} - y|$ optimization).
> 2. **Comparative metrics** (Table 2), where our method achieves superior fidelity (lowest $\ell_{rec}$ and $M_{<}$ ratios):
>
> **Table 2:** Reconstruction metrics. _mask_y_: $M$ ratio; _mask_l_: $M_{<}$ ratio; _loss_rec_: $\ell_{rec}$.
>
> |Method|mask_y ratio|mask_l ratio|loss_rec|
> |---|---|---|---|
> |patch_d|0.3558|0.0807|0.0087|
> |patch_i|0.4378|0.0780|0.0125|
> |linear|0.4198|0.3198|0.0318|
> |sparse|0.4708|0.0170|0.0008|
> |**ours**|**0.3837**|**0.0141**|**0.0002**|

---

> > ### Comment · Reviewer_wXrM · 2025-08-05
> >
> > The author's rebuttal addressed all of my concerns. I changed my current scores and support this paper acceptance.

---

> ### Author Response · Authors · 2025-08-05
> **Kind Follow-up on Our Rebuttal - Additional Feedback Appreciated**
>
> Dear Reviewer wXrM,
>
> Thank you for your thoughtful review and for taking the time to consider our rebuttal. To ensure the final manuscript reaches its full potential, we kindly ask if you could kindly:
> - share any remaining concerns or suggestions regarding our rebuttal, as this will allow us to address them effectively during the discussion phase.
> - let us know if any additional questions or clarifications are needed, so we can provide the necessary responses or experiments in a timely manner before the final deadline.
>
> We truly value your insights and are here to provide any further information you may need.
>
> Best regards,
>
> Authors of Submission 22548

---

### Official Review · Reviewer_mY2H · 2025-07-02

**Clarity:** 4
**Significance:** 4
**Originality:** 4
**Rating:** 5
**Confidence:** 5

**Summary:**

This paper focuses on the important issue of performance degradation caused by low-quality labels. The authors innovatively propose to re-label time series datasets by replacing original labels with pseudo-labels generated by reconstruction networks. Specifically, the authors put a prediction network and a reconstruction network into their proposed structure with three training objectives among prediction, raw label, and reconstructed label. Additionally, the authors introduce Self-Correction with Adaptive Mask (SCAM), using loss landscape sharpness to identify overfitting components and dynamically replace raw labels with pseudo-labels. Moreover, the authors present Spectral Norm Regularization (SNR) as a training normalization strategy to constrain the gradient of the prediction network and smooth the loss landscape.

**Questions:**

1. In equation 4, the SCAM mask strategy relies on the empirical decomposition of loss functions. Is it possible to use theories such as PAC-Bayes to clarify the correction boundary of pseudo-labels?
2. The reconstruction network is computationally heavy. Can knowledge distillation or neural architecture search be used to lower the computational resource consumption of the reconstruction network? For example, how much performance degradation will occur if we replace dense convolution layers with one-dimensional sparse convolution layers?
3. The binary mask M may lose transition state information. Can a Soft Mask or uncertainty weighting be designed to achieve continuous label correction?

**Ethical Concerns:**

["NO or VERY MINOR ethics concerns only"]

**Limitations:**

yes

**Paper Formatting Concerns:**

I have not noticed important formatting issues.

**Quality:**

4

**Strengths And Weaknesses:**

The proposed framework in this paper makes use of the self-supervising philosophy to reduce noisy labels dependence. Also, the adaptive character of SCAM is based on solid mathematical optimization heuristics (loss landscapes and overfitting), leading to high-quality toxic label filtering. Meanwhile, the authors conduct detailed analysis and ablation studies, demonstrating that their method is effective and convincing.

The authors concatenate representations from multiple stacked Convolution layers and put the concatenated representation into an FFN layer for multi-resolution reconstruction. Such a Convolution design usually causes high computational resources and relies heavily on hyperparameter choosing. In this paper, the authors use a four-layered convolutional network with kernel sizes 1,2,4,8, and the experiments on benchmark datasets confirm the feasibility of such parameter choosing. However, the authors do not include parameter analysis of the reconstruction network in their analysis. If the model’s performance is easily affected by the reconstruction network parameters, the proposed method is less attractive in realistic applications, as it heavily relies on the developer’s experience. Meanwhile, the analysis in the supplementary materials shows that adding SCAM to current backbones costs additional computational resources and time in a large proportion with adding a small amount of parameters (around 30k). This might indicate that the reconstruction network is comparably heavy and time-consuming. It would be very meaningful to look into how necessary the design of the current reconstruction network is in future works.

---

> ### Author Rebuttal · Authors · 2025-07-31
>
> **Q1: Correction Boundary of Pseudo-Labels**
>
> > We acknowledge the unfeasibility of static statistical methods and provide posterior correlation proof that is also theoretically backed.
>
> **(1) Definition of the Boundary/Mask in Current Methods**
>
> As briefly explained in Section 3.3, we observed a clear tendency toward overfitting when adding a new reconstruction loss ($\ell_{rec}$) to the supervised loss ($\ell_{tar}$). The addition has two key effects on SSL:
>
> 1. The loss reaches a lower optimum.
> 2. Overfitting occurs, preventing convergence to the lowest optimum.
>
> This observation suggests the existence of two opposing components, which motivates the derivation of **Equation 4**. We reformulate the problem as a binary classification problem. The loss components are the objects with the loss statistics of training samples as attributes. The "label" refers to whether a loss component contributes to overfitting.
>
> **(2) Complexity in Label Acquisition**
>
> While Bayesian or PCA analysis could help decide the boundary for the binary classification. The ground-truth labels are not determined by observable attributes. The labels—quantified by the overfitting metric ($\ell_{test} - \ell_{train}$)—are accessible after backward-propagation on the combined loss. Thus, we cannot decide whether **a single component** causes overfitting.
>
> **(3) Empirical Analysis: Proof of the Current Boundary from a Correlation Perspective**
>
> A more practical approach is to verify classification results (mask $M$) in an end-to-end manner. Below, we provide statistical evidence:
>
> By applying mask $M$ to $\ell_{rec}$ and $\ell_{tar}$, we obtain:
>
> - $\ell_{rec} \odot M$, $\ell_{rec} \odot \overline{M}$, $\ell_{tar} \odot M$, $\ell_{tar} \odot \overline{M}$
>     (denoted as **rec_pos, rec_neg, tar_pos, tar_neg**)
>
> Using Pearson Correlation, we evaluate how training loss correlates with test performance. A negative score indicates that the training loss negatively impacts test performance (i.e., overfitting). We focus on correlations with two target test losses, i.e. the overall performance metric.
>
> **Table 1**: Train-Test Loss Correlation (Epoch 1, 50, 100)
>
> ||tar_pos_test (1,50,100)|tar_neg_test (1, 50, 100)|
> |---|---|---|
> |**tar_pos_train**|0.000221, -0.10482, -0.101608|0.009486, -0.05257, -0.08058|
> |**tar_neg_train**|-0.02193, 0.110645, 0.033152|-0.02061, 0.060247, 0.13079|
> |**rec_pos_train**|-0.05054, -0.13014, 0.333769|0.052278, 0.014121, 0.037747|
> |**rec_neg_train**|-0.04216, 0.012024, 0.266016|0.029819, 0.088595, 0.130759|
>
> A clear trend emerges: **tar_pos_train** exhibits decreasing correlation scores over time, suggesting tar_pos_train as the contributing factor of overfitting.
>
> **Table 2: Trend of Train-Test Loss Correlation**
> _(Trend = (end correlation - begin correlation) / total epochs)_
>
> |correlations|Trend (corr/epoch) |
> |---|---|
> |tar_pos_train / tar_pos_test|-4.58e-5|
> |tar_pos_train / tar_neg_test|-13.4e-5|
> |tar_neg_train / tar_pos_test|9.57e-5|
> |tar_neg_train / tar_neg_test|7.50e-5|
> |rec_pos_train / tar_pos_test|39.2e-5|
> |rec_pos_train / tar_neg_test|14.5e-5|
> |rec_neg_train / tar_pos_test|50.9e-5|
> |rec_neg_train / tar_neg_test|37.2e-5|
>
> **(4) Conclusion**
>
> We adopt some **posterior statistical analysis** to support our conclusion. We acknowledge the value of determining the boundary on **a priori** and will discuss this further in the **Limitations/Future Work** section of the camera-ready version.
>
> **Q2: Computational Efficiency Optimization of the Reconstruction Network**
>
> > We provide updated efficiency results, which corrected a bug causing overestimation of costs. Further, we explore different architectures that trade performance for better efficiency, along with detailed component analysis.
>
> **(1) Additional Computation Costs Are Only Incurred During Training**
>
> The reconstruction network acts as a **scaffold** during training, meaning the predictor operates independently at inference time without requiring the reconstruction network.  No additional costs will be incurred in actual deployment.
>
> **(2) Correction of Efficiency Analysis for the Reconstruction Network**
>
> **Table 6 in Appendix G.4** initially overestimated the additional computational cost due to a bug in our codebase, which inadvertently doubled both the cost and parameter counts.
>
> **Root Cause:**
> The issue stemmed from the use of `tensordict.nn.EnsembleModule`. We patched the official PyTorch implementation in `src/models/modules/ensemble.py` to fix a bug in parameter registration for `TensorDictModule`. The original implementation created duplicate parameter copies instead of using the correct ones for computation.
>
> **Table 3:** Corrected efficiency benchmark.
>
> |Benchmark|Total Memory (MB)|Raw Memory (MB)|Additional Memory (MB)|Additional Params (K)|
> |---|---|---|---|---|
> |ETT (7 channels) MLP|30.31|18.15|12.16|2.7|
> |ETT iTrans|620.8|71.9|548.9|2.7|
> |Electricity (321 channels) MLP|36.72|22.23|14.49|2.7|
> |Electricity iTrans|739.43|181.47|557.96|2.7|
>
>  **(3) Optional designs for different performance-efficiency trade-offs**
>
> To study the trade-offs in reconstruction network design, we propose an **embedding-decoder abstraction**, where:
> - **Embeddings** convert time series to vectors.
> - **Decoders** reconstruct vectors back to time series.
>
> This framework allows us to analyze the cost of the following alternatives:
>
> 1. **Patch_d / patch_i**: Patch embeddings with dependent/independent linear decoders (Appendix Figure 11).
> 2. **Linear**: Vanilla MLP (linear embedding + decoder).
> 3. **Conv-sparse**: Inspired by SparseTSF, uses multi-stride convolutions for sparse embeddings.
> 4. **Ours (Conv-FFN)**: Pyramidal convolution (Figure 9), our final choice.
>
>  **(4) Detailed Component Analysis**
>
> - The memory for different components refers to peak memory usage in backpropagation of the corresponding module.
> - The **predictor memory** reflects peak usage since its loss computation depends on the reconstruction loss.
>
> **Table 4:** Component analysis on ETT.
>
> | method  | embedding mem (MB) | decoder mem (MB) | predictor/peak mem (MB) | additional memory (MB) | Backbone memory (MB) | additional params (K) |
> |---|---|---|---|---|---|---|
> |patch_d|78.13|97.51|111.33|39.43|71.9|149.4|
> |patch_i|76.77|96.01|109.82|37.92|71.9|6.3|
> |linear|65.92|72.21|85.44|13.54|71.9|37.2|
> |sparse|2683.1|2927.61|2940.84|2868.94|71.9|86.2|
> |ours|363.06|607.57|620.8|548.9|71.9|2.7|
>
> **Table 5:** Component analysis on Electricity.
>
> | method | embedding mem (MB) | decoder mem (MB) | predictor/peak mem (MB) | additional memory (MB) | raw memory (MB) | additional params (K) |
> |---|---|---|---|---|---|---|
> |patch_d|81.38|100.32|228.47|47|181.47|149.4|
> |patch_i|80.01|98.82|227.59|46.12|181.47|6.3|
> |linear|69.6|75.89|204.08|22.61|181.47|37.2|
> |sparse|724.03| 2968.54|3097.31|2915.84|181.47|86.2|
> |ours|366.21|610.72|739.43|557.96|181.47|2.7|
>
> **Key Observations**
> - **Channel count is the primary factor** influencing memory overhead. which can be alleviated by channel sampling.
> - **Efficiency-performance trade-offs**: Some alternatives (e.g., `patch_i`, `linear`) offer marginal efficiency gains at slight performance costs (see **Response to Reviewer b9qu Q2** for performance details).
>
> **(5) Conclusions**
> - We corrected the initial memory overestimation due to a bug.
> - Multiple architectural variants were explored to optimize efficiency.
> - As concluded in **Appendix G.4**, **channel sampling [1]** remains the most effective way to control reconstruction costs—aligning with our claim that reconstruction networks do **not** bottleneck TSF model scalability.
>
> _[1] iTransformer: Inverted Transformers Are Effective for Time Series Forecasting_
>
> **Q3: Soft Mask for Continuous Label Correction**
> > We verify the performance of a simple Tanh-based softmask that upderperforms. There is room for more delicate softmasks left for future studies.
>
> **(1) Types of soft mask**
> The suggestion to explore soft mask designs is indeed insightful. In machine learning, "soft" masks typically refer to differentiable gating mechanisms, such as:
> - **Softmax** (for multi-category routing)
> - **Tanh** (for binary gating)
> In our method, softening the mask involves reformulating the original binary mask definitions:
> 1. Original hard mask:
>     $M=\mathbb{I}[(\tilde{y} - \hat{y})(\tilde{y} -y) >0]$
>
>     $M_{<}=\mathbb{I}[|\tilde{y}-\hat{y}| - |\tilde{y} - y|<0]$
>     where $\mathbb{I}[\cdot]$ is the indicator function.
> 2. Proposed soft mask (using Tanh):
>     $M=\text{Tanh}[(\tilde{y} - \hat{y})(\tilde{y} -y)]$
>
>     $M_{<} = \text{Tanh} [|\tilde{y}- y| - |\tilde{y} - \hat{y}|]$
> This remaps mask values from $\{0, 1\}$ to $(0, 1)$, enabling continuous gradient flow.
>
> **(2) Experimental Results**
>
> We evaluated soft vs. hard masks across multiple datasets and architectures:
>
> **Table 6:** Performance comparisons of soft and hard masks.
>
> |Method| ETTh1| ETTm1|Weather|
> |---|---|---|---|
> |MLP+scam+Soft |0.376|0.344|0.207|
> |MLP+scam+Hard|0.373|0.319|0.176|
> |iTrans+scam+Soft|0.390|0.337|0.200|
> |iTrans+scam+Hard|0.373|0.315|0.173|
>
> **Key Findings**
> 1. **Hard masks consistently outperform soft masks** across all benchmarks.
>     - We hypothesize this is because hard masks enforce **focused optimization** on effective loss components, whereas soft masks dilute gradients by assigning partial weights to suboptimal terms.
> 2. **Tanh as a baseline soft mask**:
>     - While our experiments used Tanh (a standard choice for binary gating), other soft masking functions (e.g., Sigmoid, Learned thresholds) could be explored.
> 3. **Extension to multi-category masks**:
>     - For scenarios with **multiple reconstructions**, the mask could be generalized to a multi-category version (e.g., using Softmax to route losses from different reconstructions).
>
> We will add these discussions to the Limitations or Future Work section in the camera-ready version. We appreciate your suggestions very much and hope our responses can address some of your concerns.

---

> > ### Comment · Reviewer_mY2H · 2025-08-05
> >
> > Thanks for the authors' detailed rebuttal. All of my concerns are addressed comprehensively.

---

### Official Review · Reviewer_b9qu · 2025-07-03

**Clarity:** 3
**Significance:** 3
**Originality:** 3
**Rating:** 4
**Confidence:** 3

**Summary:**

This paper proposes a self-supervised method called SCAM, which improves time series forecasting by replacing overfitted raw labels with pseudo-labels from a reconstruction network, guided by an adaptive mask. It also uses spectral norm regularization to stabilize training. Experiments show consistent performance improvements.

**Questions:**

No

**Ethical Concerns:**

["NO or VERY MINOR ethics concerns only"]

**Final Justification:**

The authors address my concerns during the rebuttal period.

**Limitations:**

yes

**Quality:**

3

**Strengths And Weaknesses:**

Strengths:

**1**: Introducing self-supervised reconstruction into time series label generation is a promising idea, and the proposed SCAM method shows practical application potential.

**2**: The method is evaluated on various mainstream TSF models and multiple datasets, demonstrating good adaptability and performance improvement.

Weaknesses:

**1**: Although the method outperforms standard supervised training, it lacks in-depth comparison with existing self-supervised reconstruction or label denoising approaches, such as Denoising Label Learning and Mixup for time series.

**2**: The CNN+FFN architecture used for reconstruction is relatively simple, and the paper does not explore whether more advanced designs could lead to further performance gains.

**3**: Although the paper mentions that SNR helps mitigate attention collapse in Transformers, it lacks systematic empirical analysis on why SNR is effective only on linear layers and how the application points of SNR are selected.

**4**: Although the method is tested on multiple datasets, it remains limited to conventional TSF tasks and has not been validated in more complex scenarios, such as industrial sensor anomaly detection.

---

> ### Author Rebuttal · Authors · 2025-07-31
>
> **Q1: In-depth comparison with existing reconstruction or label denoising approaches**
>
> > **Summary of Below:** We address the lack of direct self-supervised baselines for continuous-label time-series forecasting (TSF) by adapting MixUp and LatentMixUp, demonstrating their potential while showing that our raw method SCAM is competitive, supported by empirical results.
>
> **(1) Baseline Selection.**
> We extensively explored SSL methods, including _Denoising Label Learning_ and _MixUp_. However, these are either:
> - Operate on vastly different data modalities (making them unsuitable for time series), or
> - Work with discrete labels, which are essentially different from continuous labels in forecasting, requiring additional adaptation.
>
> Our work is the **first** to construct **continuous pseudo-labels** for TSF by integrating an auxiliary self-supervised reconstruction task. **Thus, no off-the-shelf baselines are directly applicable to our scenario.**
> To address this, we adapted _MixUp_ and _LatentMixUp_ [1,2] as self-supervised baselines for TSF—demonstrating SSL without a reconstruction network.
>
> [1] _mixup: Beyond Empirical Risk Minimization_
>
> [2] _Embarrassingly Simple MixUp for Time-series_
>
> **(2) Adaptations of SSL Baselines.**
> The core idea of _MixUp_ is linear interpolation:
> $$\begin{aligned} \tilde{x} &= \lambda x_{1} +(1-\lambda)x_{2} \\\\ \tilde{y} &= \lambda y_{1} +(1-\lambda)y_{2}, \end{aligned}$$
> where a predictor $f$ is optimized on $||f(\tilde{x})-\tilde{y}||$. _LatentMixUp_ extends _MixUp_ to latent space, which replaces $x$ with $h$.
>
> ***Adaptation 1:*** For **discrete labels**, linearly combining one-hot vectors is reasonable due to sparsity. However, randomly interpolating time series can be meaningless. We find that selecting $x_1$​ and​ $x_2$ based on **implicit periodicity** (using priors from [3]) works effectively within _MixUp_.
>
> [3] _CycleNet: Enhancing Time Series Forecasting through Modeling Periodic Patterns_ (NeurIPS 2024 spotlight)
>
> ***Adaptation 2:*** Following standard classification models, we treat the TSF model’s hidden state as the input to the final linear projector (a common design in TSF baselines). Other alternatives were tested but underperformed.
>
> Following _MixUp++_ [2], we use multiple $\lambda$ values ([0.3, 0.5, 0.7]) instead of a single one.
>
> **(3) Empirical Results and Insights.**
> Experiments follow Section 4 and Appendix G (rerun with minor adjustments; results may slightly differ from the paper).
>
> **Table 1:** Performance of _MixUp_ and _LatentMixUp_ with MLP and iTransformer.
>
> | | ETTh1 | ETTh2| ETTm1| ETTm2 |
> | ------------------ | --------- | --------- | --------- | --------- |
> | method  | MSE | MSE| MSE | MSE |
> | MLP+MixUp | 0.376| 0.285 | 0.331 | 0.177 |
> | MLP+LatentMixUp | 0.381 | 0.289 | 0.339| 0.183 |
> | **MLP****+scam**   | **0.373** | **0.280** | 0.**319** | **0.175** |
> | iTrans+MixUp| **0.371** | 0.296 | 0.315 | 0.187  |
> | iTrans+LatentMixup | 0.412 | 0.295  | 0.396 | 0.197|
> | **iTrans+scam**  | 0.373 | **0.293** | **0.315** | **0.179** |
>
>
> Insights from Table 1:
> - **SSL is promising for TSF.** To our knowledge, _MixUp_ (with intra-period mixing) has not been applied to TSF before. Our adaptation shows its potential.
> - **Non-latent SSL outperforms latent SSL.** Both _MixUp_ and our method (_SCAM_) manipulate raw time series, which outperforms _LatentMixUp_.
>
> Adapting SSL baselines, although potentially performant, requires **additional effort** without on-the-shelf implementations. We will expand on this in the paper and explore it further in future work. We hope these efforts address your concerns.
>
> **Q2: Exploration of other advanced designs of reconstruction networks**
>
> > **Summary of Below:** We focus on the SSL framework rather than reconstruction architecture design, and the empirical results show minimal performance differences between feasible alternatives.
>
>  **(1) Reasons for Limited Exploration of Reconstruction Network Designs**.
> 1. **Focus on SSL Paradigm**: Our work emphasizes the novel self-supervised learning (SSL) framework rather than architectural innovation or feature engineering.
> 2. **Task-Specific Performance**: Predictor model performance does not directly translate to reconstruction tasks.
> 3. **Minimal performance gaps**: Early experiments showed minimal performance gaps across architectures.
>
> In summary, the reconstruction network’s role is solely to enable our SSL paradigm. Extensive architectural exploration would deviate from our core contribution and lack prior work for reference.
>
> **(2) Explored Reconstruction Network Designs**.
> We evaluated multiple architectures, among which feasible ones are listed below, categorized by **embedding** and **decoder** components:
>
> **Table 3**: Design Choices for Reconstruction Networks
>
> | **Embedding** | **Decoder**       | **Abbr.** |
> | ------------- | ----------------- | --------- |
> | Patch-emb     | Patch-dependent   | `patch_d` |
> | Patch-emb | Patch-independent | `patch_i` |
> | Linear  | Linear-dependent  | `linear`  |
> | Conv-sparse   | Point-wise        | `sparse`  |
> | Conv-ffn| Point-wise        | `ours` |
>
> 1. **Patch_d/patch_i**: Patch embeddings with dependent/independent linear decoders (Appendix Figure 11).
> 2. **Linear**: Vanilla MLP (linear embedding + decoder).
> 3. **Conv-sparse**: Inspired by SparseTSF [4], uses multi-stride convolutions for sparse embeddings.
> 4. **Ours (Conv-FFN)**: Pyramidal convolution (Figure 9), our final choice.
>
> [4] *SparseTSF: Modeling Long-term Time Series Forecasting with 1k Parameters* (ICML 2024 Oral)
>
> **Table 4**: Performance Comparison
>
> |Method | ETTh1 | ETTh2| ETTm1| ETTm2|
> | ------- | --------- | --------- | --------- | --------- |
> | | MSE  | MSE | MSE | MSE  | MSE |
> |patch_d| 0.374| **0.280** | 0.320| 0.176 |
> |patch_i| **0.373** | **0.280** | 0.322 | **0.175** |
> |linear| 0.380 | 0.287 | 0.324 | 0.177|
> |sparse| 0.375| 0.281| **0.318** | 0.176 |
> |ours| **0.373** | **0.280** | 0.319  | **0.175** |
>
> The performance gaps between these options are not very significant, which explains why we do not focus much on the topic.
>
> **(3) Conclusion**
>
> Reconstruction network design is secondary to our SSL framework. While feasible alternatives exist, their performance differences are negligible, and optimizing them falls outside our paper’s scope. We appreciate the reviewer’s feedback and hope this clarification addresses the concerns.
>
> **Q3: systematic empirical analysis of SNR**
>
> > **Summary of Below:** The ineffectiveness of SNR on self-attention is a conclusion drawn from [5]. We focus more on the empirical analysis of the detailed effective field for SNR on common components of TSF models.
>
> **(1) The existing empirical study regarding SNR**
>
> The application of SNR or SNR-like (spectral normalization operation) to alleviate overfitting is a classical approach. However, very few studies have discussed its application specifically in Time Series Forecasting. [5] first studies this topic, and has concluded the ineffectiveness of pure SNR on attention matrices with empirical studies already.
>
> However, their conclusions are limited to the two constraints:
>
> 1. They study channel-wise transformers particularly. The overfitting phenomenon is attributed to channel-wise attention.
> 2. They verify the ineffectiveness of Spectral Normalization on self-attention only following previous works [6].
>
> **(2) The empirical study in our paper**
>
> As pointed out in our paper (in Appendix E.3), we observe that the overfitting problem, measured by sharpness metric, does not solely exist in attention-relevant component (Encoder in Figures 16,17), but also exists in linear layers. We therefore add SNR to linear layers in TSF models for empirical verifications. As described in Appendix E.3, by adding SNR, the overfitting issue is alleviated for all three components.
>
> We also conduct empirical analysis on where the SNR should be applied (Figure 18) supported by actual performances. These results may be model-dependent and therefore only serve as a reference.
>
> The ablation study regarding SNR is also included in Section 4.2. We conclude that the alleviation of overfitting is especially crucial for SCAM, and using SNR alone may cause sub-optimal performance sometimes.
>
> **(3) Conclusion**
>
> We do not intend to lay much emphasis on the root theory of SNR and extend the previous analysis in [5], and supplement their conclusion about the feasible field of the SNR method. Our empirical results show that SNR is not a silver bullet but works well with SCAM.
>
> _[5] SAMformer: Unlocking the Potential of Transformers in Time Series Forecasting with Sharpness-Aware Minimization and Channel-Wise Attention_
>
> _[6] Stabilizing Transformer Training by Preventing Attention Entropy Collapse_
>
> **Q4: Application in more complex scenarios like industrial sensor anomaly detection**
>
> Our work restricts the topic to Time Series Forecasting only, due to the great gap between the self-supervision in classification and forecasting. The fundamental essence of our method, which is distinguishing noise, however, does relate to anomaly detection. The mask value, under reasonable adjustments can be utilized as noise indicators, or augment training data for external anomaly detectors. We will seriously consider such an opportunity to extend our work to other tasks, including anomaly detection and will add such discussion in the section about future works in the camera-ready version.
>
> > *Looking forward to further discussions.* All experiment results displayed in the rebuttal are selected due to the total character constraint; we are pleased to provide comprehensive statistics in the following phase if requested. Additionally, further concerns about the content of our response or other topics are welcome in the discussion phase. We appreciate your valuable advice and look forward to your reply.

---

> > ### Author Response · Authors · 2025-08-05
> > **Kind Follow-up on Our Rebuttal - Additional Feedback Appreciated**
> >
> > Dear Reviewer b9qu,
> >
> > Thank you for your thoughtful review and for taking the time to consider our rebuttal. To ensure the final manuscript reaches its full potential, we kindly ask if you could kindly:
> > - share any remaining concerns or suggestions regarding our rebuttal, as this will allow us to address them effectively during the discussion phase.
> > - let us know if any additional questions or clarifications are needed, so we can provide the necessary responses or experiments in a timely manner before the final deadline.
> >
> > We truly value your insights and are here to provide any further information you may need.
> >
> > Best regards,
> >
> > Authors of Submission 22548

---

> > ### Comment · Reviewer_b9qu · 2025-08-05
> >
> > Thanks for your detailed reply and I increase my score.

---

### Comment · Area_Chair_pqJE · 2025-08-05

Dear Reviewers:

The Author-Reviewer Discussion Period will remain open until August 8 (AoE).

Your active participation during this phase is essential. Please:

- Read the author responses and other reviews.

- Engage constructively with authors to clarify any concerns.

Thanks to those who have already begun their discussions, and to all of you for your hard work with these reviews.

AC

---

### Author Response · Authors · 2025-08-09

Dear Reviewers,

We sincerely thank all reviewers for your unanimous feedback that our rebuttal has fully addressed your concerns. We are deeply grateful for this positive consensus. In the final version of the paper, we will incorporate every clarification and the additional experimental evidence discussed, ensuring that the improvements are clearly and thoroughly reflected.

Best regards,

Authors of Submission 22548

---

### Decision · Program_Chairs · 2025-09-17

**Decision:**

Accept (spotlight)

**Comment:**

This paper proposes a self-supervised method called SCAM, which improves time series forecasting by replacing overfitted raw labels with pseudo-labels from a reconstruction network, guided by an adaptive mask, and employs spectral norm regularization (SNR) to stabilize training. The idea is considered novel and well-motivated (mY2H, GL6M), practically effective across multiple datasets and backbone models (b9qu, wXrM), and theoretically grounded with clear visualizations (GL6M, mY2H). Although some weaknesses remain, including limited comparisons with existing self-supervised/denoising methods (b9qu), computational overhead of the reconstruction network (mY2H, wXrM), the authors have addressed the issues in their responses.